# Relationship between Batch Size and Number of Steps Needed for Nonconvex Optimization of Stochastic Gradient Descent using Armijo Line Search

## Abstract

Stochastic gradient descent (SGD) is the simplest deep learning optimizer with which to train deep neural networks. While SGD can use various learning rates, such as constant or diminishing rates, the previous numerical results showed that SGD performs better than other deep learning optimizers using when it uses learning rates given by line search methods. In this paper, we perform a convergence analysis on SGD with a learning rate given by an Armijo line search for nonconvex optimization. The analysis indicates that the upper bound of the expectation of the squared norm of the full gradient becomes small when the number of steps and the batch size are large. Next, we show that, for SGD with the Armijo-line-search learning rate, the number of steps needed for nonconvex optimization is a monotone decreasing convex function of the batch size; that is, the number of steps needed for nonconvex optimization decreases as the batch size increases. Furthermore, we show that the stochastic first-order oracle (SFO) complexity, which is the stochastic gradient computation cost, is a convex function of the batch size; that is, there exists a critical batch size that minimizes the SFO complexity. Finally, we provide numerical results that support our theoretical results. The numerical results indicate that the number of steps needed for training deep neural networks decreases as the batch size increases and that there exist the critical batch sizes that can be estimated from the theoretical results.

## 1 Introduction

### 1.1 Background

Nonconvex optimization is useful for training deep neural networks, since the loss functions called the expected risk and empirical risk are nonconvex and they need only be minimized in order to find the model parameters. Deep-learning optimizers have been presented for minimizing the loss functions. The simplest one is stochastic gradient descent (SGD) (Robbins & Monro, 1951; Zinkevich, 2003; Nemirovski et al., 2009; Ghadimi & Lan, 2012; 2013) and there are numerous theoretical analyses on using SGD for nonconvex optimization (Jain et al., 2018; Vaswani et al., 2019; Fehrman et al., 2020; Chen et al., 2020; Scaman & Malherbe, 2020; Loizou et al., 2021). Variants have also been presented, such as momentum methods (Polyak, 1964; Nesterov, 1983) and adaptive methods including Adaptive Gradient (AdaGrad) (Duchi et al., 2011), Root Mean Square Propagation (RMSProp) (Tieleman & Hinton, 2012), Adaptive Moment Estimation (Adam) (Kingma & Ba, 2015), Adaptive Mean Square Gradient (AMSGrad) (Reddi et al., 2018), and Adam with decoupled weight decay (AdamW) (Loshchilov & Hutter, 2019). SGD and its variants are useful for training not only deep neural networks but also generative adversarial networks (Heusel et al., 2017; Naganuma & Iiduka, 2023; Sato & Iiduka, 2023).

The performance of deep-learning optimizers for nonconvex optimization depends on the batch size. The previous numerical results in (Shallue et al., 2019) and (Zhang et al., 2019) have shown that the number of steps $K$ needed to train a deep neural network halves for each doubling of the batch size $b$ and that there is a region of diminishing returns beyond the *critical batch size* $b^\star$. This fact can be expressed as follows: there

is a positive number $C$ such that $N := Kb \approx C$ for $b \leq b^\star$ and $N := Kb \geq C$ for $b \geq b^\star$. The deep neural network model uses $b$ gradients of the loss functions per step. Hence, when $K$ is the number of steps required to train a deep neural network, the model has a stochastic gradient computation cost of $Kb$. We will define the *stochastic first-order oracle (SFO) complexity* (Iiduka, 2022; Sato & Iiduka, 2023) of a deep-learning optimizer to be $N := Kb$. From the previous numerical results in (Shallue et al., 2019) and (Zhang et al., 2019), the SFO complexity is minimized at a critical batch size $b^\star$ and there are diminishing returns once the batch size exceeds $b^\star$. Therefore, it is desirable to use the critical batch size when minimizing the SFO complexity of the deep-learning optimizer.

Not only a batch size but also a learning rate affects the performance of deep-learning optimizers for nonconvex optimization. A performance measure of a deep-learning optimizer generating a sequence $(\boldsymbol{\theta}_k)_{k \in \mathbb{N}}$ is the expectation of the squared norm of the gradient of a nonconvex loss function $f$, denoted by $\mathbb{E}[\|\nabla f(\boldsymbol{\theta}_k)\|^2]$. If this performance measure becomes small when the number of steps $k$ is large, the deep-learning optimizer approximates a local minimizer of $f$. For example, let us consider the problem of minimizing a smooth function $f$ (see Section 2.1 for the definition of smoothness). Here, SGD using a constant learning rate $\alpha = O(\frac{1}{L})$ satisfies $\min_{k \in [K]} \mathbb{E}\left[\|\nabla f(\boldsymbol{\theta}_k)\|^2\right] = O(\frac{1}{K} + \frac{\alpha}{b})$, where $L$ is the Lipschitz constant of $\nabla f$, $b$ is the batch size, and $[K] := \{1, 2, \ldots, K\}$ (see also Table 1). Moreover, SGD using a learning rate satisfying the Armijo condition was presented in (Vaswani et al., 2019). The *Armijo line search* (Nocedal & Wright, 2006, Chapter 3.1) is a standard method for finding an appropriate learning rate $\alpha_k$ giving a sufficient decrease in $f$, i.e., $f(\boldsymbol{\theta}_{k+1}) < f(\boldsymbol{\theta}_k)$ (see Section 2.3.1 for the definition of the Armijo condition).

## 1.2 Motivation

The numerical results in (Vaswani et al., 2019) indicated that the Armijo-line-search learning rate is superior to using a constant learning rate when using SGD to train deep neural networks in the sense of minimizing the training loss functions and improving test accuracy. Motivated by the useful numerical results in (Vaswani et al., 2019), we decided to perform convergence analyses on SGD with the Armijo-line-search learning rate for nonconvex optimization in deep neural networks.

Theorem 3 in (Vaswani et al., 2019) is a convergence analysis of SGD with the Armijo-line-search learning rate for nonconvex optimization under a strong growth condition that implies the interpolation property. Here, let $f \colon \mathbb{R}^d \to \mathbb{R}$ be an empirical risk defined by $f(\boldsymbol{\theta}) := \frac{1}{n} \sum_{i \in [n]} f_i(\boldsymbol{\theta})$, where $n$ is the number of training data and $f_i \colon \mathbb{R}^d \to \mathbb{R}$ is a loss function corresponding to the $i$-th training data $z_i$. We say that $f$ has the interpolation property if $\nabla f(\boldsymbol{\theta}) = \mathbf{0}$ implies $\nabla f_i(\boldsymbol{\theta}) = \mathbf{0}$ $(i \in [n])$. The interpolation property holds for optimization of a linear model with the squared hinge loss for binary classification on linearly separable data (Vaswani et al., 2019, Section 2). However, the interpolation condition would be unrealistic for deep neural networks, since their loss functions are nonconvex. The motivation behind this work is thus to show that SGD with the Armijo-line-search learning rate can solve nonconvex optimization problems in deep neural networks.

As indicated the second paragraph in Section 1.1, the batch size has a significant effect on the performance of SGD. Hence, in accordance with the first motivation stated above, we decided to investigate appropriate batch sizes for SGD with the Armijo-line-search learning rate. In particular, we were interested in verifying whether a critical batch size $b^\star$ minimizing the SFO complexity $N$ exists for training deep neural networks with SGD using the Armijo condition in theory and in practice. This is because the previous studies in (Shallue et al., 2019; Zhang et al., 2019; Iiduka, 2022; Sato & Iiduka, 2023) showed the existence of critical batch sizes for training deep neural networks or generative adversarial networks with optimizers with constant or diminishing learning rates and without Armijo-line-search learning rates.

We are also interested in estimating critical batch sizes before implementing SGD with the Armijo-line-search learning rate. The previous results in (Iiduka, 2022; Sato & Iiduka, 2023) showed that, for optimizers using constant learning rates, the critical batch sizes determined from numerical results are close to the theoretically estimated sizes. Motivated by the results in (Iiduka, 2022; Sato & Iiduka, 2023), we sought to verify whether, for SGD with the Armijo-line-search learning rate, the measured critical batch sizes are close to the batch sizes estimated from theoretical results.

### 1.3 Contribution

#### 1.3.1 Convergence analysis of SGD with Armijo-line-search learning rates

The first contribution of this paper is to present a convergence analysis of SGD with Armijo-line-search learning rates for general nonconvex optimization (Theorem 3.1); in particular, it is shown that SGD with this rate $\alpha_k$ satisfies that, for all $K \geq 1$,

$$
\min_{k \in [0:K-1]} \mathbb{E}\left[\|\nabla f(\boldsymbol{\theta}_k)\|^2\right] \leq \underbrace{\overbrace{\frac{2(f(\boldsymbol{\theta}_0) - f_*)}{(2 - L_n\overline{\alpha})\underline{\alpha}}}^{C_1} \frac{1}{K}}_{B(\boldsymbol{\theta}_0, K)} + \underbrace{\overbrace{\frac{\overline{\alpha}\sigma^2}{(2 - L_n\overline{\alpha})\underline{\alpha}}}^{C_2} \frac{1}{b}}_{V(\sigma^2, b)}, \tag{1}
$$

where the parameters are defined in Table 1 (see also Theorem 3.1). The inequality (1) indicates that the upper bound of the performance measure $\min_{k \in [0:K-1]} \mathbb{E}[\|\nabla f(\boldsymbol{\theta}_k)\|^2]$ that consists of a bias term $B(\boldsymbol{\theta}_0, K)$ and variance term $V(\sigma^2, b)$ becomes small when the number of steps $K$ is large and the batch size $b$ is large. Therefore, it is desirable to set $K$ large and $b$ large so that Algorithm 1 will approximate a local minimizer of $f$.

The essential lemma to proving (1) is the guarantee of the existence of a lower bound on the learning rates satisfying the Armijo condition (Lemma 2.1). Although, in general, learning rates satisfying the Armijo condition do not have any lower bound (Lemma 2.1(i)), the corresponding learning rates computed by a backtracking line search (Algorithm 2) have a lower bound (Lemma 2.1(ii)). In addition, the descent lemma (i.e., $f(\boldsymbol{y}) \leq f(\boldsymbol{x}) + \langle \nabla f(\boldsymbol{x}), \boldsymbol{y} - \boldsymbol{x} \rangle + \frac{L_n}{2} \|\boldsymbol{y} - \boldsymbol{x}\|^2$ $(\boldsymbol{x}, \boldsymbol{y} \in \mathbb{R}^d)$) holds from the smoothness condition on $f$. Thus, we can prove (1) by using the existence of a lower bound on the learning rates satisfying the Armijo condition and the descent lemma (see Appendix A.2 for details of the proof of Theorem 3.1).

Table 1: Relationship between batch size $b$ and number of steps $K$ to achieve an $\epsilon$–approximation defined by $\min_{k \in [0:K-1]} \mathbb{E}[\|\nabla f(\boldsymbol{\theta}_k)\|^2] \leq \frac{C_1}{K} + \frac{C_2}{b} = \epsilon^2$ for SGD with a constant learning rate $\alpha \in (0, \frac{2}{L_n})$ and for SGD with the Armijo-line-search learning rate $\alpha_k \in [\underline{\alpha}, \overline{\alpha}]$ ($[0:K-1] := \{0, 1, \ldots, K-1\}$, $f := \frac{1}{n} \sum_{i \in [n]} f_i$ is bounded below by $f_*$, $L_i$ is the Lipschitz constant of $\nabla f_i$, $L_n := \frac{1}{n} \sum_{i \in [n]} L_i$, and $\sigma^2$ is the upper bound of the variance of the stochastic gradient)

| Learning Rate | Upper Bound $\frac{C_1}{K} + \frac{C_2}{b}$ | | Steps $K$ | SFO $N$ | Critical Batch $b^\star$ |
|---|---|---|---|---|---|
| Constant $\alpha \in \left(0, \frac{2}{L_n}\right)$ | $C_1$ | $\frac{2(f(\boldsymbol{\theta}_0) - f_*)}{(2 - L_n\alpha)\alpha}$ | $K = \frac{C_1 b}{\epsilon^2 b - C_2}$ | $N = \frac{C_1 b^2}{\epsilon^2 b - C_2}$ | $b^\star = \frac{2C_2}{\epsilon^2}$ |
| | $C_2$ | $\frac{L_n\alpha\sigma^2}{2 - L_n\alpha}$ | | | |
| Armijo $(c, \delta \in (0,1))$ | $C_1$ | $\frac{2(f(\boldsymbol{\theta}_0) - f_*)}{(2 - L_n\overline{\alpha})\underline{\alpha}}$ | $K = \frac{C_1 b}{\epsilon^2 b - C_2}$ | $N = \frac{C_1 b^2}{\epsilon^2 b - C_2}$ | $b^\star = \frac{2C_2}{\epsilon^2}$ |
| | $C_2$ | $\frac{\overline{\alpha}\sigma^2}{(2 - L_n\overline{\alpha})\underline{\alpha}}$ | | | |

To show the merit of SGD with the Armijo-line-search learning rate, we compare an implementation using this rate with one using a constant learning rate (see, e.g., (Scaman & Malherbe, 2020, Section 4) for convergence analyses of SGD with constant learning rates). In this case, we need to set a constant learning rate $\alpha \in (0, \frac{2}{L_n})$ depending on the Lipschitz constant $L_n$ of $\nabla f$ (see also Table 1). However, computing $L_n$ is NP-hard (Virmaux & Scaman, 2018), so it would be unrealistic to set a constant learning rate depending on $L_n$ before implementing SGD. Meanwhile, we need to set $c, \delta \in (0,1)$ in order to use SGD with the Armijo-line-search learning rate. We would like to emphasize here that we can choose *any* $c, \delta \in (0,1)$ to implement SGD. That is, for any $c, \delta \in (0,1)$, there exists a learning rate satisfying the Armijo condition (see (11) for the definition of the Armijo condition) and it can be found by conducting a simple backtracking line search (Algorithm 2) instead of performing a complicated computation such as of the Lipschitz constant of $\nabla f$ (Lemma 2.1).

### 1.3.2 Steps needed for $\epsilon$–approximation of SGD with Armijo line-search-learning rates

The previous results in (Shallue et al., 2019; Zhang et al., 2019; Iiduka, 2022; Sato & Iiduka, 2023) indicated that, for optimizers, the number of steps $K$ needed to train a deep neural network or generative adversarial networks decreases as the batch size increases. The second contribution of this paper is to show that, for SGD with the Armijo-line-search learning rate, the number of steps $K$ needed for nonconvex optimization decreases as the batch size increases. Let us consider the case in which the right-hand side of (1) is equal to $\epsilon^2$, where $\epsilon > 0$ is the precision. Then, $K$ is a rational function defined for a batch size $b$ by

$$K = K(b) = \frac{C_1 b}{\epsilon^2 b - C_2}, \tag{2}$$

where $C_1$ and $C_2$ are the positive constants defined in (1) (see also Table 1). We can easily show that $K$ defined above is a monotone decreasing and convex function with respect to $b$ (Theorem 3.2). Accordingly, the number of steps needed for nonconvex optimization decreases as the batch size increases.

### 1.3.3 Critical batch size minimizing SFO complexity of SGD with Armijo-line-search learning rates

Using $K$ defined by (2) above, we can further define the SFO complexity $N$ of SGD with Armijo-line-search learning rates (see also Table 1):

$$N = Kb = K(b)b = \frac{C_1 b^2}{\epsilon^2 b - C_2}. \tag{3}$$

We can easily show that $N$ is convex with respect to $b$ and that a global minimizer

$$b^\star = \frac{2C_2}{\epsilon^2} = \frac{2\overline{\alpha}\sigma^2}{(2 - L_n\overline{\alpha})\underline{\alpha}\epsilon^2} \tag{4}$$

exists for it (Theorem 3.3). Accordingly, there is a critical batch size $b^\star$ at which $N$ is minimized.

Here, we compare the number of steps $K_{\mathrm{C}}$ and the SFO complexity $N_{\mathrm{C}}$ for SGD using a constant learning rate $\alpha$ with $K_{\mathrm{A}}$ and $N_{\mathrm{A}}$ for SGD using the Armijo-line-search learning rate $\alpha_k$ ($\in [\underline{\alpha}, \overline{\alpha}]$). Let $C_{1,\mathrm{C}}$ (resp. $C_{2,\mathrm{C}}$) be $C_1$ (resp. $C_2$) in Table 1 for SGD using a constant learning rate and let $C_{1,\mathrm{A}}$ (resp. $C_{2,\mathrm{A}}$) be $C_1$ (resp. $C_2$) in Table 1 for SGD using the Armijo-line-search learning rate. We have that

$$
\begin{aligned}
C_{1,\mathrm{A}} < C_{1,\mathrm{C}} \text{ iff } \underline{\alpha} &> \frac{2 - L_n\alpha}{2 - L_n\overline{\alpha}}\alpha, \\
C_{2,\mathrm{A}} < C_{2,\mathrm{C}} \text{ iff } \frac{\overline{\alpha}}{\underline{\alpha}} &< \frac{\sigma_{\mathrm{C}}^2(2 - L_n\overline{\alpha})}{\sigma_{\mathrm{A}}^2(2 - L_n\alpha)}L_n\alpha,
\end{aligned}
\tag{5}
$$

where $\sigma_{\mathrm{C}}^2$ (resp. $\sigma_{\mathrm{A}}^2$) denotes the upper bound of the variance of the stochastic gradient for SGD using a constant learning rate $\alpha$ (resp. the Armijo-line-search learning rate). If (5) holds, then SGD using the Armijo-line-search learning rate converges faster than SGD using a constant learning rate in the sense that

$$\frac{C_{1,\mathrm{A}} b}{\epsilon^2 b - C_{2,\mathrm{A}}} = K_{\mathrm{A}} < K_{\mathrm{C}} = \frac{C_{1,\mathrm{C}} b}{\epsilon^2 b - C_{2,\mathrm{C}}} \text{ and } \frac{C_{1,\mathrm{A}} b^2}{\epsilon^2 b - C_{2,\mathrm{A}}} = N_{\mathrm{A}} < N_{\mathrm{C}} = \frac{C_{1,\mathrm{C}} b^2}{\epsilon^2 b - C_{2,\mathrm{C}}}.$$

It would be difficult to check exactly that (5) holds before implementing SGD, since (5) involves unknown parameters, such as $L_n = \frac{1}{n}\sum_{i \in [n]} L_i$, $\sigma_{\mathrm{C}}^2$, and $\sigma_{\mathrm{A}}^2$. However, it can be expected that (5) holds, since it is known empirically (Vaswani et al., 2019, Figure 5) that the relationship between the Armijo-line-search learning rate $\alpha_k$ and a constant learning rate $\alpha$ is $\alpha < \alpha_k \in [\underline{\alpha}, \overline{\alpha}]$ (Section 3.3 provides the derivation of condition (5)).

### 1.3.4 Numerical results supporting our theoretical results

The numerical results in (Vaswani et al., 2019) showed that SGD with the Armijo-line-search learning rate performs better than other optimizers in training deep neural networks. Hence, we sought to verify whether

the numerical results match our theoretical results (Sections 1.3.1, 1.3.2, and 1.3.3). We trained residual networks (ResNets) on the CIFAR-10 and CIFAR-100 datasets and a two-hidden-layer multi-layer perceptron (MLP) on the MNIST dataset. We numerically found that increasing the batch size $b$ decreases the number of steps $K$ needed to achieve high training accuracies and that there are critical batch sizes minimizing the SFO complexities. We also estimated batch sizes using (4) for the critical batch size $b^\star$ and compared them with ones determined from the numerical results. We found that the estimated batch sizes are close to the ones determined from the numerical results. To verify whether SGD using the Armijo-line-search learning rate performs better than SGD using a constant learning rate (see the discussion in condition (5)), we numerically compared SGD using the Armijo-line-search learning rate with not only SGD using a constant learning rate but also variants of SGD, such as the momentum method, Adam, AdamW, and RMSProp. We found that SGD using the Armijo-line-search learning rate and the critical batch size performs better than other optimizers in the sense of minimizing the number of steps and the SFO complexities needed to achieve high training accuracies (Section 4).

## 2 Mathematical Preliminaries

### 2.1 Definitions

Let $\mathbb{N}$ be the set of nonnegative integers, $[n] := \{1, 2, \ldots, n\}$ for $n \geq 1$, and $[0 : n] := \{0, 1, \ldots, n\}$ for $n \geq 0$. Let $\mathbb{R}^d$ be a $d$–dimensional Euclidean space with inner product $\langle \cdot, \cdot \rangle$ inducing the norm $\| \cdot \|$.

Let $f \colon \mathbb{R}^d \to \mathbb{R}$ be continuously differentiable. We denote the gradient of $f$ by $\nabla f \colon \mathbb{R}^d \to \mathbb{R}^d$. Let $L > 0$. $f \colon \mathbb{R}^d \to \mathbb{R}$ is said to be $L$–smooth if $\nabla f \colon \mathbb{R}^d \to \mathbb{R}^d$ is $L$–Lipschitz continuous, i.e., for all $\boldsymbol{x}, \boldsymbol{y} \in \mathbb{R}^d$, $\|\nabla f(\boldsymbol{x}) - \nabla f(\boldsymbol{y})\| \leq L\|\boldsymbol{x} - \boldsymbol{y}\|$. When $f \colon \mathbb{R}^d \to \mathbb{R}$ is $L$–smooth, the following inequality, called the descent lemma (Beck, 2017, Lemma 5.7), holds: for all $\boldsymbol{x}, \boldsymbol{y} \in \mathbb{R}^d$, $f(\boldsymbol{y}) \leq f(\boldsymbol{x}) + \langle \nabla f(\boldsymbol{x}), \boldsymbol{y} - \boldsymbol{x} \rangle + \frac{L}{2}\|\boldsymbol{y} - \boldsymbol{x}\|^2$. Let $f_* \in \mathbb{R}$. $f \colon \mathbb{R}^d \to \mathbb{R}$ is said to be bounded below by $f_*$ if, for all $\boldsymbol{x} \in \mathbb{R}^d$, $f(\boldsymbol{x}) \geq f_*$.

### 2.2 Assumptions and problem

Given a parameter $\boldsymbol{\theta} \in \mathbb{R}^d$ and a data point $z$ in a data domain $Z$, a machine learning model provides a prediction whose quality can be measured by a differentiable nonconvex loss function $f(\boldsymbol{\theta}; z)$. We aim to minimize the empirical average loss defined for all $\boldsymbol{\theta} \in \mathbb{R}^d$ by

$$f(\boldsymbol{\theta}) = \frac{1}{n} \sum_{i \in [n]} f(\boldsymbol{\theta}; z_i) = \frac{1}{n} \sum_{i \in [n]} f_i(\boldsymbol{\theta}),$$

where $S = (z_1, z_2, \ldots, z_n)$ denotes the training set and $f_i(\cdot) := f(\cdot; z_i)$ denotes the loss function corresponding to the $i$-th training data $z_i$.

This paper considers the following smooth nonconvex optimization problem.

**Problem 2.1** *Suppose that $f_i \colon \mathbb{R}^d \to \mathbb{R}$ ($i \in [n]$) is $L_i$–smooth and bounded below by $f_{i,*}$. Then,*

$$\text{minimize } f(\boldsymbol{\theta}) := \frac{1}{n} \sum_{i \in [n]} f_i(\boldsymbol{\theta}) \text{ subject to } \boldsymbol{\theta} \in \mathbb{R}^d.$$

We assume that a stochastic first-order oracle (SFO) exists such that, for a given $\boldsymbol{\theta} \in \mathbb{R}^d$, it returns a stochastic gradient $\mathsf{G}_\xi(\boldsymbol{\theta})$ of the function $f$, where a random variable $\xi$ is supported on a finite/an infinite set $\Xi$ (i.e., $\text{supp}(\xi) = \{x \in \Xi \colon \xi(x) \neq 0\}$) independently of $\boldsymbol{\theta}$. We make the following standard assumptions.

**Assumption 2.1**

(A1) Let $(\boldsymbol{\theta}_k)_{k \in \mathbb{N}} \subset \mathbb{R}^d$ be the sequence generated by SGD. For each iteration $k$,

$$\mathbb{E}_{\xi_k}[\mathsf{G}_{\xi_k}(\boldsymbol{\theta}_k)] = \nabla f(\boldsymbol{\theta}_k), \tag{6}$$

where $\xi_0, \xi_1, \ldots$ are independent samples and the random variable $\xi_k$ is independent of $(\boldsymbol{\theta}_l)_{l=0}^k$. There exists a nonnegative constant $\sigma^2$ such that

$$\mathbb{E}_{\xi_k} \left[ \|\mathsf{G}_{\xi_k}(\boldsymbol{\theta}_k) - \nabla f(\boldsymbol{\theta}_k)\|^2 \right] \leq \sigma^2. \tag{7}$$

(A2) For each iteration $k$, SGD samples a batch $B_k$ of size $b$ independently of $k$ and estimates the full gradient $\nabla f$ as

$$\nabla f_{B_k}(\boldsymbol{\theta}_k) := \frac{1}{b} \sum_{i \in [b]} \mathsf{G}_{\xi_{k,i}}(\boldsymbol{\theta}_k) = \frac{1}{b} \sum_{i \in [b]} \nabla f_{\xi_{k,i}}(\boldsymbol{\theta}_k),$$

where $\xi_{k,i}$ is a random variable generated by the $i$-th sampling in the $k$-th iteration.

### 2.3 Stochastic gradient descent using Armijo line search

#### 2.3.1 Armijo condition

Suppose that $f \colon \mathbb{R}^d \to \mathbb{R}$ is continuously differentiable. We would like to find a stationary point $\boldsymbol{\theta}^\star \in \mathbb{R}^d$ such that $\nabla f(\boldsymbol{\theta}^\star) = \mathbf{0}$ by using an iterative method defined by

$$\boldsymbol{\theta}_{k+1} := \boldsymbol{\theta}_k + \alpha_k \boldsymbol{d}_k, \tag{8}$$

where $\alpha_k > 0$ is the step size (called a learning rate in the machine learning field) and $\boldsymbol{d}_k \in \mathbb{R}^d$ is the search direction. Various methods can be used depending on the search direction $\boldsymbol{d}_k$. For example, the method (8) with $\boldsymbol{d}_k := -\nabla f(\boldsymbol{\theta}_k)$ is gradient descent, while the method (8) with $\boldsymbol{d}_k := -\nabla f(\boldsymbol{\theta}_k) + \beta_{k-1} \boldsymbol{d}_{k-1}$, where $\beta_k \geq 0$, is the conjugate gradient method. If we define $\boldsymbol{d}_k$ (e.g., $\boldsymbol{d}_k := -\nabla f(\boldsymbol{\theta}_k)$), it is desirable to set $\alpha_k^\star$ satisfying

$$f(\boldsymbol{\theta}_k + \alpha_k^\star \boldsymbol{d}_k) = \min_{\alpha > 0} f(\boldsymbol{\theta}_k + \alpha \boldsymbol{d}_k). \tag{9}$$

The step size $\alpha_k^\star$ defined by (9) can be easily computed when $f$ is quadratic and convex. However, for a general nonconvex function $f$, it is difficult to compute the step size $\alpha_k^\star$ in (9) exactly. Here, we can use the *Armijo condition* for finding an appropriate step size $\alpha_k$: Let $c \in (0, 1)$. We would like to find $\alpha_k > 0$ such that

$$f(\boldsymbol{\theta}_k + \alpha_k \boldsymbol{d}_k) \leq f(\boldsymbol{\theta}_k) + c\alpha_k \langle \nabla f(\boldsymbol{\theta}_k), \boldsymbol{d}_k \rangle. \tag{10}$$

When $\boldsymbol{d}_k$ satisfies the descent property defined by $\langle \nabla f(\boldsymbol{\theta}_k), \boldsymbol{d}_k \rangle < 0$ (e.g., gradient descent using $\boldsymbol{d}_k := -\nabla f(\boldsymbol{\theta}_k)$ has the property such that $\langle \nabla f(\boldsymbol{\theta}_k), \boldsymbol{d}_k \rangle = -\|\nabla f(\boldsymbol{\theta}_k)\|^2 < 0$), the Armijo condition ensures that $f(\boldsymbol{\theta}_{k+1}) = f(\boldsymbol{\theta}_k + \alpha_k \boldsymbol{d}_k) < f(\boldsymbol{\theta}_k)$. Accordingly, $\alpha_k$ satisfying the Armijo condition (10) is appropriate in the sense of minimizing $f$.

The existence of step sizes satisfying the Armijo condition (10) is guaranteed.

**Proposition 2.1** (Nocedal & Wright, 2006, Lemma 3.1) *Let $f \colon \mathbb{R}^d \to \mathbb{R}$ be continuously differentiable. Let $\boldsymbol{\theta}_k \in \mathbb{R}^d$ and let $\boldsymbol{d}_k$ ($\neq \mathbf{0}$) have the descent property defined by $\langle \nabla f(\boldsymbol{\theta}_k), \boldsymbol{d}_k \rangle < 0$. Let $c \in (0, 1)$. Then, there exists $\gamma_k > 0$ such that, for all $\alpha_k \in (0, \gamma_k]$, the Armijo condition (10) holds.*

#### 2.3.2 Stochastic gradient descent under Armijo condition

The objective of this paper is to solve Problem 2.1 using mini-batch SGD under Assumption 2.1 defined by

$$\boldsymbol{\theta}_{k+1} = \boldsymbol{\theta}_k + \alpha_k \boldsymbol{d}_k = \boldsymbol{\theta}_k - \alpha_k \nabla f_{B_k}(\boldsymbol{\theta}_k) = \boldsymbol{\theta}_k - \frac{\alpha_k}{b} \sum_{i \in [b]} \mathsf{G}_{\xi_{k,i}}(\boldsymbol{\theta}_k),$$

where $b > 0$ is the batch size and $\alpha_k > 0$ is the learning rate. For each iteration $k$, we can use $\boldsymbol{\theta}_k$, $f_{B_k}$, and $\nabla f_{B_k}$. Hence, the Armijo condition (Vaswani et al., 2019, (1)) at the $k$-th iteration for SGD can be obtained by replacing $f$ in (10) with $f_{B_k}$ and using $\boldsymbol{d}_k = -\nabla f_{B_k}(\boldsymbol{\theta}_k)$:

$$f_{B_k}(\boldsymbol{\theta}_k - \alpha_k \nabla f_{B_k}(\boldsymbol{\theta}_k)) \leq f_{B_k}(\boldsymbol{\theta}_k) - c\alpha_k \|\nabla f_{B_k}(\boldsymbol{\theta}_k)\|^2. \tag{11}$$

The Armijo condition (11) ensures that $f_{B_k}(\boldsymbol{\theta}_{k+1}) = f_{B_k}(\boldsymbol{\theta}_k - \alpha_k \nabla f_{B_k}(\boldsymbol{\theta}_k)) < f_{B_k}(\boldsymbol{\theta}_k)$; i.e., the Armijo condition (11) is appropriate in the sense of minimizing the estimated objective function $f_{B_k}$ from the full objective function $f$. In fact, the numerical results in (Vaswani et al., 2019, Section 7) indicate that SGD using the Armijo condition (11) is superior to using other deep-learning optimizers to train deep neural networks.

Algorithm 1 is the SGD algorithm using the Armijo condition (11).

---

**Algorithm 1** Stochastic gradient descent using Armijo line search

---

**Require:** $c \in (0, 1)$ (hyperparameter), $b > 0$ (batch size), $\boldsymbol{\theta}_0 \in \mathbb{R}^d$ (initial point), $K \geq 1$ (steps)
**Ensure:** $\boldsymbol{\theta}_K \in \mathbb{R}^d$
  $k \leftarrow 0$
  **for** $k = 0, 1, \dots, K - 1$ **do**
    Compute $\alpha_k > 0$ satisfying $f_{B_k}(\boldsymbol{\theta}_k - \alpha_k \nabla f_{B_k}(\boldsymbol{\theta}_k)) \leq f_{B_k}(\boldsymbol{\theta}_k) - c\alpha_k \|\nabla f_{B_k}(\boldsymbol{\theta}_k)\|^2$ ◁ Algorithm 2
    Compute $\boldsymbol{\theta}_{k+1} = \boldsymbol{\theta}_k - \alpha_k \nabla f_{B_k}(\boldsymbol{\theta}_k)$
  **end for**

---

The search direction of Algorithm 1 is $\boldsymbol{d}_k = -\nabla f_{B_k}(\boldsymbol{\theta}_k)$ $(\neq \boldsymbol{0})$ which has the descent property defined by $\langle \nabla f_{B_k}(\boldsymbol{\theta}_k), \boldsymbol{d}_k \rangle = -\|\nabla f_{B_k}(\boldsymbol{\theta}_k)\|^2 < 0$. Hence, from Proposition 2.1, there exists a learning rate $\alpha_k \in (0, \gamma_k]$ satisfying the Armijo condition (11). Moreover, the proposition guarantees that the learning rate can be chosen to be sufficiently small, e.g., $\liminf_{k \to +\infty} \alpha_k = 0$.

The convergence analyses of Algorithm 1 use a lower bound of $\alpha_k \in (0, \gamma_k]$ satisfying the Armijo condition (11). To guarantee the existence of such a lower bound, we use the backtracking method ((Nocedal & Wright, 2006, Algorithm 3.1) and (Vaswani et al., 2019, Algorithm 2)) described in Algorithm 2.

---

**Algorithm 2** Backtracking Armijo-line-search method (Nocedal & Wright, 2006, Algorithm 3.1)

---

**Require:** $c, \delta, \frac{1}{\gamma} \in (0, 1)$ (hyperparameters), $\alpha = \gamma^{\frac{b}{n}} \alpha_{k-1}$ (initialization), $\boldsymbol{\theta}_k \in \mathbb{R}^d$, $f_{B_k} : \mathbb{R}^d \to \mathbb{R}$
**Ensure:** $\alpha_k$ satisfying $f_{B_k}(\boldsymbol{\theta}_k - \alpha_k \nabla f_{B_k}(\boldsymbol{\theta}_k)) \leq f_{B_k}(\boldsymbol{\theta}_k) - c\alpha_k \|\nabla f_{B_k}(\boldsymbol{\theta}_k)\|^2$
  **repeat**
    $\alpha \leftarrow \delta\alpha$
  **until** $f_{B_k}(\boldsymbol{\theta}_k - \alpha \nabla f_{B_k}(\boldsymbol{\theta}_k)) \leq f_{B_k}(\boldsymbol{\theta}_k) - c\alpha \|\nabla f_{B_k}(\boldsymbol{\theta}_k)\|^2$

---

The following lemma guarantees the existence of a lower bound on the learning rates computed by Algorithm 2. The proof is given in Appendix A.1.

**Lemma 2.1** *Consider Algorithm 1 under Assumption 2.1 for solving Problem 2.1. Let $\alpha_k$ be a learning rate satisfying the Armijo condition (11) (whose existence is guaranteed by Proposition 2.1), let $L_{B_k}$ be the Lipschitz constant of $\nabla f_{B_k}$, and let $L$ be the maximum value of the Lipschitz constant $L_i$ of $\nabla f_i$. Then, the following hold.*

(i) *[Counter-example of (Vaswani et al., 2019, Lemma 1)] There exists Problem 2.1 such that $\alpha_k$ does not satisfy $\min\{\frac{2(1-c)}{L_{B_k}}, \overline{\alpha}\} \leq \alpha_k$, where $\overline{\alpha}$ is an upper bound of $\alpha_k$.*

(ii) *[Lower bound on learning rate determined by backtracking line search method] If $\alpha_k$ can be computed by Algorithm 2, then there exists a lower bound of $\alpha_k$ such that $0 < \underline{\alpha} := \frac{2\delta(1-c)}{L} \leq \frac{2\delta(1-c)}{L_{B_k}} \leq \alpha_k$.*

## 3 Analysis of SGD using Armijo Line Search

### 3.1 Convergence analysis of Algorithm 1

Here, we present a convergence analysis of Algorithm 1. The proof of Theorem 3.1 is given in Appendix A.2.

**Theorem 3.1 (Upper bound of the squared norm of the full gradient)** *Consider the sequence $(\boldsymbol{\theta}_k)_{k \in \mathbb{N}}$ generated by Algorithm 1 under Assumption 2.1 for solving Problem 2.1 and suppose that the*

*learning rate $\alpha_k \in [\underline{\alpha}, \overline{\alpha}]$ is computed by Algorithm 2. Then, for all $K \geq 1$,*

$$\min_{k \in [0:K-1]} \mathbb{E}\left[\|\nabla f(\boldsymbol{\theta}_k)\|^2\right] \leq \underbrace{\overbrace{\frac{2(f(\boldsymbol{\theta}_0) - f_*)}{(2 - L_n\overline{\alpha})\underline{\alpha}}}^{C_1} \frac{1}{K}}_{B(\boldsymbol{\theta}_0, K)} + \underbrace{\overbrace{\frac{\overline{\alpha}\sigma^2}{(2 - L_n\overline{\alpha})\underline{\alpha}}}^{C_2} \frac{1}{b}}_{V(\sigma^2, b)},$$

*where $\delta, c \in (0,1)$, $L := \max_{i \in [n]} L_i$, $L_n := \frac{1}{n}\sum_{i \in [n]} L_i$ ($\leq L$), $f_* := \frac{1}{n}\sum_{i \in [n]} f_{i,*}$, $\underline{\alpha} := \frac{2\delta(1-c)}{L}$, and $\overline{\alpha} < \frac{1}{L_n}$.*

Theorem 3.1 indicates that the upper bound of the minimum value of $\mathbb{E}[\|\nabla f(\boldsymbol{\theta}_k)\|^2]$ consists of a bias term $B(\boldsymbol{\theta}_0, K)$ and variance term $V(\sigma^2, b)$. When the number of steps $K$ is large and the batch size $b$ is large, $B(\boldsymbol{\theta}_0, K)$ and $V(\sigma^2, b)$ become small. Therefore, we need to set $K$ large and $b$ large so that Algorithm 1 will approximate a local minimizer of $f$.

For the sake of convenience, we list below all assumptions considered in Theorem 3.1:

(1) [Smoothness of loss functions] $f_i \colon \mathbb{R}^d \to \mathbb{R}$ ($i \in [n]$) is $L_i$–smooth and bounded below by $f_{i,*}$ (see Problem 2.1). This implies that $f := \frac{1}{n}\sum_{i \in [n]} f_i$ is $L_n$–smooth.

(2) [Conditions of stochastic gradient] Let $\boldsymbol{\theta}_k$ be the $k$-th iteration generated by SGD and let $\mathsf{G}_{\xi_k}(\boldsymbol{\theta}_k)$ be the stochastic gradient of $f$.

    (a) [Unbiased estimator] $\mathbb{E}_{\xi_k}[\mathsf{G}_{\xi_k}(\boldsymbol{\theta}_k)] = \nabla f(\boldsymbol{\theta}_k)$ (see (6) in Assumption 2.1(A1)).

    (b) [Bounded variance] $\mathbb{E}_{\xi_k}\left[\|\mathsf{G}_{\xi_k}(\boldsymbol{\theta}_k) - \nabla f(\boldsymbol{\theta}_k)\|^2\right] \leq \sigma^2$ (see (7) in Assumption 2.1(A1)).

    (c) [Mini-batch stochastic gradient] $\nabla f_{B_k}(\boldsymbol{\theta}_k) := \frac{1}{b}\sum_{i \in [b]} \mathsf{G}_{\xi_{k,i}}(\boldsymbol{\theta}_k) = \frac{1}{b}\sum_{i \in [b]} \nabla f_{\xi_{k,i}}(\boldsymbol{\theta}_k)$ (see Assumption 2.1(A2)).

(3) [Armijo-line-search learning rate] Let $\alpha_k$ be a learning rate satisfying the Armijo condition (11) (whose existence is guaranteed by Proposition 2.1).

    (a) [Computability] We assume that $\alpha_k$ can be computed by using the backtracking Armijo-line-search method (Algorithm 2). Lemma 2.1(ii) thus guarantees the existence of a lower bound $\underline{\alpha} := \frac{2\delta(1-c)}{L}$ of $\alpha_k$. The existence of a upper bound $\overline{\alpha}$ of $\alpha_k$ is guaranteed by Proposition 2.1.

    (b) [Condition of upper bound] We assume that $\overline{\alpha} < \frac{1}{L_n}$ to ensure that $C_1$ and $C_2$ are positive (see Theorem 3.1 for the definitions of $C_1$ and $C_2$).

Here, we compare Theorem 3.1 with the convergence analysis of SGD using a constant learning rate. SGD using a constant learning rate $\alpha \in (0, \frac{2}{L_n})$ satisfies

$$\min_{k \in [0:K-1]} \mathbb{E}\left[\|\nabla f(\boldsymbol{\theta}_k)\|^2\right] \leq \frac{2(f(\boldsymbol{\theta}_0) - f_*)}{(2 - L_n\alpha)\alpha} \frac{1}{K} + \frac{L_n\alpha\sigma^2}{2 - L_n\alpha} \frac{1}{b} \tag{12}$$

(The proof of (12) is given in Appendix A.5). We need to set a constant learning rate $\alpha \in (0, \frac{2}{L_n})$ depending on the Lipschitz constant $L_n$ of $\nabla f$. However, since computing $L_n$ is NP-hard (Virmaux & Scaman, 2018), it is difficult to set $\alpha \in (0, \frac{2}{L_n})$. Meanwhile, it is sufficient to set $c, \delta \in (0, 1)$ in Algorithms 1 and 2 without computing the Lipschitz constant of $\nabla f$.

We also compare Theorem 3.1 with Theorem 3 in (Vaswani et al., 2019). Theorem 3 in (Vaswani et al., 2019) indicates that, under a strong growth condition with a constant $\rho$ (i.e., $\mathbb{E}_i[\|\nabla f_i(\boldsymbol{\theta})\|^2] \leq \rho\|\nabla f(\boldsymbol{\theta})\|^2$ ($\boldsymbol{\theta} \in \mathbb{R}^d$)) and the Armijo condition, SGD satisfies that

$$\min_{k \in [0:K-1]} \mathbb{E}\left[\|\nabla f(\boldsymbol{\theta}_k)\|^2\right] \leq \frac{f(\boldsymbol{\theta}_0) - f(\boldsymbol{\theta}^\star)}{\Delta K},$$

where $c > 1 - \frac{L}{\rho L_n}$, $\overline{\alpha} < \frac{2}{\rho L_n}$, $\Delta := (\overline{\alpha} + \frac{2(1-c)}{L}) - \rho(\overline{\alpha} - \frac{2(1-c)}{L} + L_n\overline{\alpha}^2)$, and $\boldsymbol{\theta}^\star$ is a local minimizer of $f$. Theorem 3.1 is a convergence analysis of Algorithm 1 without assuming the strong growth condition or limiting the hyperparameter $c$. Moreover, Theorem 3.1 shows that using large batch size is appropriate for SGD using the Armijo line search (Algorithm 1).

### 3.2 Steps needed for $\epsilon$–approximation

To investigate the relationship between the number of steps $K$ needed for nonconvex optimization and the batch size $b$, we consider an $\epsilon$–approximation of Algorithm 1 defined as follows:

$$\min_{k \in [0:K-1]} \mathbb{E}\left[\|\nabla f(\boldsymbol{\theta}_k)\|^2\right] \leq \epsilon^2, \tag{13}$$

where $\epsilon > 0$ is the precision.

Theorem 3.1 leads to the following theorem indicating the relationship between $b$ and the values of $K$ that achieves an $\epsilon$–approximation. The proof of Theorem 3.2 is given in Appendix A.3.

**Theorem 3.2 (Steps needed for nonconvex optimization of SGD using Armijo line search)**
*Suppose that the assumptions in Theorem 3.1 hold. Define $K \colon \mathbb{R} \to \mathbb{R}$ for all $b > \frac{C_2}{\epsilon^2}$ by*

$$K(b) = \frac{C_1 b}{\epsilon^2 b - C_2}, \tag{14}$$

*where the positive constants $C_1$ and $C_2$ are defined as in Theorem 3.1. Then, the following hold:*

(i) *[Steps needed for nonconvex optimization] $K$ defined by (14) achieves an $\epsilon$–approximation (13).*

(ii) *[Properties of the steps] $K$ defined by (14) is monotone decreasing and convex for $b > \frac{C_2}{\epsilon^2}$.*

Theorem 3.2 ensures that the number of steps $K$ needed for SGD using the Armijo line search to be an $\epsilon$–approximation is small when the batch size $b$ is large. Therefore, it is useful to set a sufficiently large batch size in the sense of minimizing the steps needed for an $\epsilon$–approximation of SGD using the Armijo line search.

We also consider setting small batch sizes, e.g., $b = 1$. From the condition of the domain of $K$, we need to satisfy

$$b = 1 > \frac{C_2}{\epsilon^2} \text{ iff } \frac{\overline{\alpha}}{\underline{\alpha}} < \frac{(2 - L_n \overline{\alpha})\epsilon^2}{\sigma^2} \tag{15}$$

to ensure the results in Theorem 3.2. If the upper bound $\overline{\alpha}$ satisfies the more restricted condition (15) than $\overline{\alpha} < \frac{1}{L_n}$, then SGD using the Armijo line search with $b = 1$ and $K(1) = \frac{C_1}{\epsilon^2 - C_2}$ is an $\epsilon$–approximation (13).

### 3.3 Critical batch size minimizing SFO complexity

The following theorem shows the existence of a critical batch size for SGD using the Armijo line search. The proof of Theorem 3.3 is given in Appendix A.4.

**Theorem 3.3 (Existence of critical batch size for SGD using Armijo line search)** *Suppose that the assumptions in Theorem 3.1 hold. Define SFO complexity $N \colon \mathbb{R} \to \mathbb{R}$ for the number of steps $K$, defined by (14), needed for an $\epsilon$–approximation (13) and for a batch size $b > \frac{C_2}{\epsilon^2}$ by*

$$N(b) = K(b)b = \frac{C_1 b^2}{\epsilon^2 b - C_2}, \tag{16}$$

*where the positive constants $C_1$ and $C_2$ are defined as in Theorem 3.1. Then, the following hold:*

(i) *[SFO complexity] $N$ defined by (16) is convex for $b > \frac{C_2}{\epsilon^2}$.*

(ii) *[Critical batch size] There exists a critical batch size*

$$b^\star = \frac{2C_2}{\epsilon^2} = \frac{2\overline{\alpha}\sigma^2}{(2 - L_n \overline{\alpha})\underline{\alpha}\epsilon^2} \tag{17}$$

*such that $b^\star$ minimizes the SFO complexity (16).*

(iii) [Upper bound on critical batch size] *The critical batch size $b^\star$ defined by (17) satisfies*

$$b^\star \leq \frac{\overline{\alpha}\sigma^2}{\{\underline{\alpha} - \delta(1-c)\overline{\alpha}\}\epsilon^2}.$$  (18)

Here, we compare the number of steps $K_\mathrm{C}$ and the SFO complexity $N_\mathrm{C}$ for SGD using a constant learning rate $\alpha$ with $K_\mathrm{A}$ and $N_\mathrm{A}$ for SGD using the Armijo-line-search learning rate $\alpha_k$ ($\in [\underline{\alpha}, \overline{\alpha}]$). Let $C_{1,\mathrm{C}}$ (resp. $C_{2,\mathrm{C}}$) be $C_1$ (resp. $C_2$) in Table 1 (see also (12)) for SGD using a constant learning rate and let $C_{1,\mathrm{A}}$ (resp. $C_{2,\mathrm{A}}$) be $C_1$ (resp. $C_2$) in Table 1 (see also Theorem 3.1) for SGD using the Armijo-line-search learning rate. We have that

$$\frac{2(f(\boldsymbol{\theta}_0) - f_*)}{(2 - L_n\overline{\alpha})\underline{\alpha}} = C_{1,\mathrm{A}} < C_{1,\mathrm{C}} = \frac{2(f(\boldsymbol{\theta}_0) - f_*)}{(2 - L_n\alpha)\alpha} \text{ iff } \underline{\alpha} > \frac{2 - L_n\alpha}{2 - L_n\overline{\alpha}}\alpha.$$  (19)

Moreover,

$$\frac{\overline{\alpha}\sigma_\mathrm{A}^2}{(2 - L_n\overline{\alpha})\underline{\alpha}} = C_{2,\mathrm{A}} < C_{2,\mathrm{C}} = \frac{L_n\alpha\sigma_\mathrm{C}^2}{2 - L_n\alpha} \text{ iff } \frac{\overline{\alpha}}{\underline{\alpha}} < \frac{\sigma_\mathrm{C}^2(2 - L_n\overline{\alpha})}{\sigma_\mathrm{A}^2(2 - L_n\alpha)}L_n\alpha,$$  (20)

where $\sigma_\mathrm{C}^2$ (resp. $\sigma_\mathrm{A}^2$) denotes the upper bound of the variance of the stochastic gradient for SGD using a constant learning rate $\alpha$ (resp. the Armijo-line-search learning rate). If (19) and (20) hold, then SGD using the Armijo-line-search learning rate converges faster than SGD using a constant learning rate in the sense that

$$\frac{C_{1,\mathrm{A}}b}{\epsilon^2 b - C_{2,\mathrm{A}}} = K_\mathrm{A} < K_\mathrm{C} = \frac{C_{1,\mathrm{C}}b}{\epsilon^2 b - C_{2,\mathrm{C}}} \text{ and } \frac{C_{1,\mathrm{A}}b^2}{\epsilon^2 b - C_{2,\mathrm{A}}} = N_\mathrm{A} < N_\mathrm{C} = \frac{C_{1,\mathrm{C}}b^2}{\epsilon^2 b - C_{2,\mathrm{C}}}.$$

It can be expected that (19) and (20) hold, since it is known empirically (Vaswani et al., 2019, Figure 5) that the relationship between the Armijo-line-search learning rate $\alpha_k$ and constant learning rate $\alpha$ is $\alpha < \alpha_k$. The next section numerically compares SGD using the Armijo-line-search learning rate with not only SGD using a constant learning rate but also variants of SGD and examines the performance of SGD using the Armijo-line-search learning rate.

The previous results in (Shallue et al., 2019; Zhang et al., 2019; Iiduka, 2022) show that, for deep-learning optimizers, there are critical batch sizes at which the SFO complexities are minimized. We are interested in verifying whether a critical batch size exists for SGD using the Armijo line search. Theorem 3.3(iii) indicates that an upper bound on the critical batch size can be obtained from some hyperparameters. Accordingly, we would like to estimate the critical batch size using the upper bound (18). Therefore, the next section numerically examines the relationship between the batch size $b$ and the number of steps $K$ needed for nonconvex optimization and the relationship between $b$ and the SFO complexity $N$ to check if there is a critical batch size $b^\star$ minimizing $N$ and if the critical batch size $b^\star$ can be estimated from our theoretical results.

## 4 Numerical Results

We verified whether numerical results match our theoretical results (Theorems 3.2 and 3.3), that is, the relationship between $K$ and $b$ and the relationship between $N$ and $b$ for Algorithm 1. We also compared the performance of Algorithm 1 with the performances of other optimizers, such as SGD with a constant learning rate (SGD), momentum method (Momentum), Adam, AdamW, and RMSProp. The learning rate and hyperparameters of the five optimizers used in each experiment were determined on the basis of a grid search.

The metrics were the number of steps $K$ and the SFO complexity $N = Kb$ indicating that the training accuracy is higher than a certain score. We used Algorithm 1 with the Armijo-line-search learning rate computed by Algorithm 2 with $\gamma = 2$, $\delta = 0.5$, $\overline{\alpha} = 10$ (see https://github.com/IssamLaradji/sls for the setting of parameters), and various values of $c$. The stopping condition was 200 epochs. The experimental environment consisted of eight NVIDIA DGX A100 GPUs and two Dual AMD Rome7742 2.25-GHz, 128-Core CPUs. The software environment was Python 3.8.2, PyTorch 1.6.0, and CUDA 11.6. The code is available at https://anonymous.4open.science/r/armijo_linesearch-C1C3.

## 4.1 Training ResNet and MLP on the CIFAR-10, CIFAR-100, and MNIST datasets

We trained ResNet-34 on the CIFAR-10 dataset ($n = 50000$). Figure 1 plots the number of steps needed for the training accuracy to be more than 0.99 for Algorithm 1 versus batch size. It can be seen that Algorithm 1 decreases the number of steps as the batch size increases. Figure 2 plots the SFO complexities of Algorithm 1 versus the batch size. It indicates that there are critical batch sizes that minimize the SFO complexities.

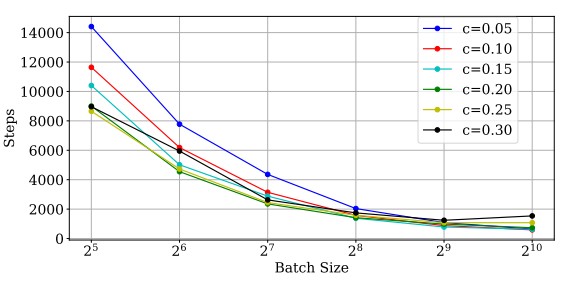

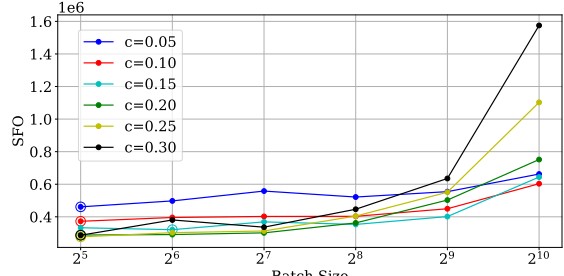

Figure 1: Number of steps for Algorithm 1 versus batch size needed to train ResNet-34 on CIFAR-10

Figure 2: SFO complexity for Algorithm 1 versus batch size needed to train ResNet-34 on CIFAR-10 (The double-circle symbol denotes the measured critical batch size)

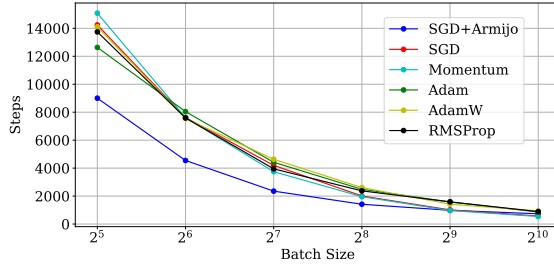

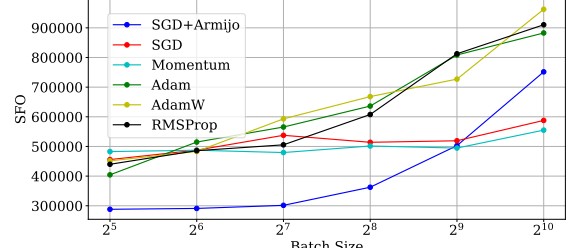

Figure 3: Number of steps for Algorithm 1 with $c = 0.20$ and variants of SGD versus batch size needed to train ResNet-34 on CIFAR-10

Figure 4: SFO complexity for Algorithm 1 with $c = 0.20$ and variants of SGD versus batch size needed to train ResNet-34 on CIFAR-10

Figures 3 and 4 compare the performance of Algorithm 1 with $c = 0.20$ with those of variants of SGD. The figures indicate that, when the batch sizes are from $2^5$ to $2^9$, SGD+Armijo (Algorithm 1) performs better than the other optimizers. In particular, the SFO complexity of SGD+Armijo (Algorithm 1) using $c = 0.20$ and the critical batch size ($b^\star = 2^5$) is the smallest of other optimizers for any batch size.

We also considered the case of training ResNet-34 on the CIFAR-100 dataset ($n = 50000$). Figure 5 plots the number of steps needed for the training accuracy to be more than 0.99 for Algorithm 1 versus the batch size, and Figure 6 plots the SFO complexities of Algorithm 1 versus the batch size. As in Figures 1 and 2, these figures show that Algorithm 1 decreases the number of steps as the batch size increases and there are critical batch sizes that minimize the SFO complexities.

Figures 7 and 8 compare the performance of Algorithm 1 with $c = 0.25$ with those of variants of SGD. The figures indicate that, when the batch sizes are from $2^5$ to $2^9$, SGD+Armijo (Algorithm 1) performs well. In particular, the SFO complexities of SGD and SGD+Armijo (Algorithm 1) using $c = 0.25$ and the critical batch size ($b^\star = 2^6$) are smaller than the SFO complexities of the other optimizers for any batch size. However, Figure 8 indicates that the SFO complexity of SGD+Armijo (Algorithm 1) increases once the batch size exceeds the critical value, as promised in Theorem 3.3.

We trained a two-hidden-layer MLP with widths of 512 and 256 on the MNIST dataset ($n = 60000$). Figure 9 plots the number of steps needed for the training accuracy to be more than 0.97 for Algorithm 1 versus

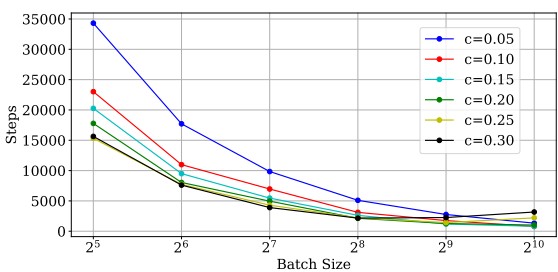

Figure 5: Number of steps for Algorithm 1 versus batch size needed to train ResNet-34 on CIFAR-100

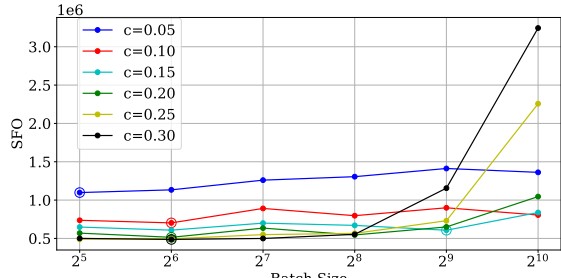

Figure 6: SFO complexity for Algorithm 1 versus batch size needed to train ResNet-34 on CIFAR-100 (The double-circle symbol denotes the measured critical batch size)

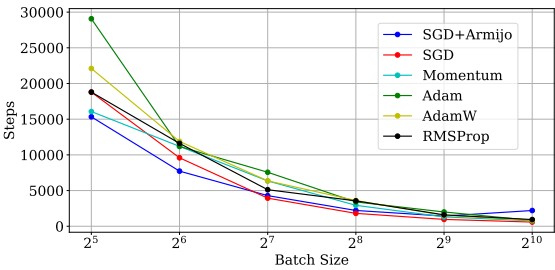

Figure 7: Number of steps for Algorithm 1 with $c = 0.25$ and variants of SGD versus batch size needed to train ResNet-34 on CIFAR-100

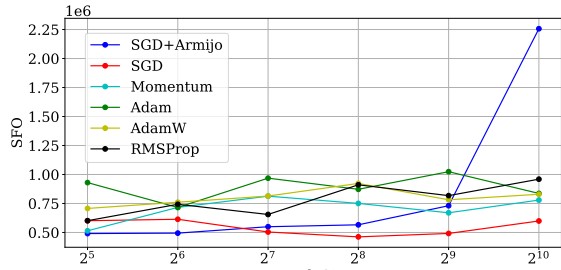

Figure 8: SFO complexity for Algorithm 1 with $c = 0.25$ and variants of SGD versus batch size needed to train ResNet-34 on CIFAR-100

the batch size, and Figure 10 plots the SFO complexities of Algorithm 1 versus the batch size. As in Figures 1, 2, 5, and 6, these figures show that Algorithm 1 decreases the number of steps as the batch size increases and there are critical batch sizes that minimize the SFO complexities.

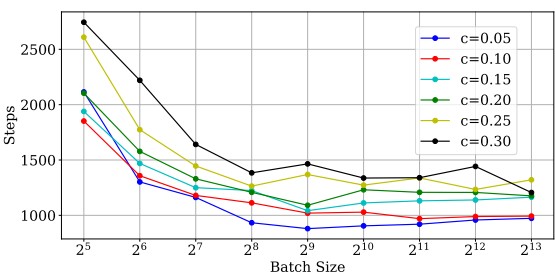

Figure 9: Number of steps for Algorithm 1 versus batch size needed to train MLP on MNIST

Figure 10: SFO complexity for Algorithm 1 versus batch size needed to train MLP on MNIST (The double-circle symbol denotes the measured critical batch size)

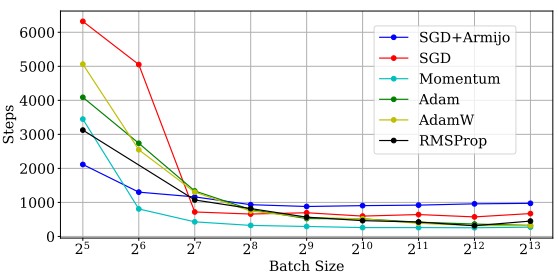
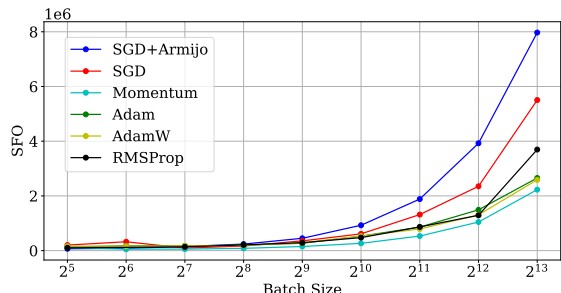

Figure 11: Number of steps for Algorithm 1 with $c = 0.05$ and variants of SGD versus batch size needed to train MLP on MNIST

Figure 12: SFO complexity for Algorithm 1 with $c = 0.05$ and variants of SGD versus batch size needed to train MLP on MNIST

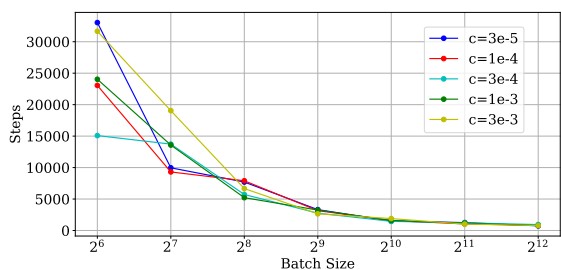
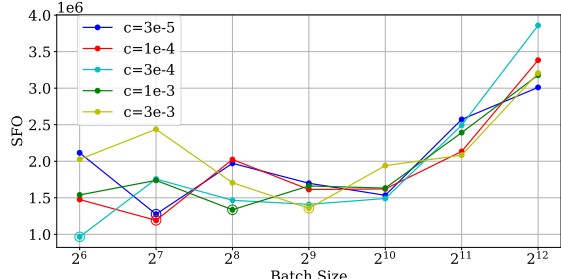

Figure 13: Number of steps for Algorithm 1 versus batch size needed to train Wide ResNet-50-2 on CIFAR-10

Figure 14: SFO complexity for Algorithm 1 versus batch size needed to train Wide ResNet-50-2 on CIFAR-10 (The double-circle symbol denotes the measured critical batch size)

Figures 11 and 12 compare the performance of Algorithm 1 with $c = 0.05$ with those of variants of SGD. The figures indicate that SGD+Armijo (Algorithm 1) using $c = 0.05$ and the critical batch size ($b^\star = 2^5$) performs better than the other optimizers in the sense of minimizing the SFO complexity. However, as was seen in Figure 8, Figure 12 indicates that the SFO complexity of SGD+Armijo (Algorithm 1) increases once the batch size exceeds the critical value.

We trained Wide ResNet-50-2 (Zagoruyko & Komodakis, 2017) on the CIFAR-10 dataset ($n = 50000$). Figure 13 plots the number of steps needed for the training accuracy to be more than 0.99 for Algorithm 1 versus the batch size. It can be seen that Algorithm 1 decreases the number of steps as the batch size increases. Figure 14 plots the SFO complexities of Algorithm 1 versus the batch size. It indicates that there are critical batch sizes that minimize the SFO complexities.

Figures 15 and 16 compare the performance of Algorithm 1 with $c = 3 \times 10^{-4}$ with those of variants of SGD. The figures indicate that SGD+Armijo (Algorithm 1) using $c = 3 \times 10^{-4}$ and the critical batch size ($b^\star = 2^6$) performs better than the other optimizers in the sense of minimizing the SFO complexity. However, as seen in Figures 8 and 12, Figure 16 indicates that the SFO complexity of SGD+Armijo (Algorithm 1) increases once the batch size exceeds the critical value.

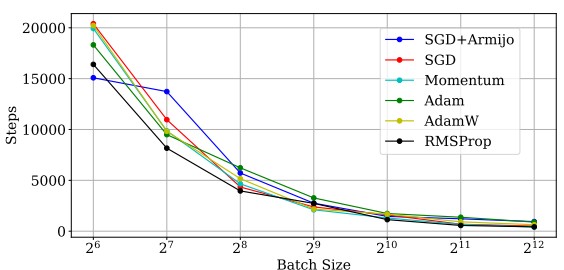
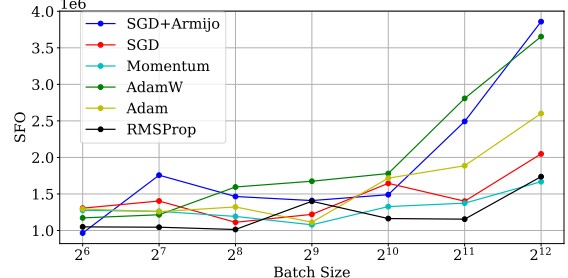

Figure 15: Number of steps for Algorithm 1 with $c = 3 \times 10^{-4}$ and variants of SGD versus batch size needed to train Wide ResNet-50-2 on CIFAR-10

Figure 16: SFO complexity for Algorithm 1 with $c = 3 \times 10^{-4}$ and variants of SGD versus batch size needed to train Wide ResNet-50-2 on CIFAR-10

Therefore, we can conclude that Algorithm 1 using the critical batch size $b^\star$ ($\in \{2^5, 2^6\}$) performs better than other optimizers using any batch size in the sense of minimizing the SFO complexities needed to achieve high training accuracies.

## 4.2 Estimation of critical batch size

We estimated the critical batch size by using Theorem 3.3(iii) and the ideas presented in (Iiduka, 2022) and (Sato & Iiduka, 2023). We used Algorithm 1 with $c = 0.05$ for training ResNet-34 on the CIFAR-100 dataset (Figures 5 and 6). Theorem 3.3(iii) indicates that the upper bound of the critical batch size involves the unknown value $\sigma^2$. We checked that the Armijo-line-search learning rates for Algorithm 1 with $c = 0.05$ are about 10 (see also (Vaswani et al., 2019, Figure 5 (Left))). Hence, we used $\underline{\alpha} \approx \overline{\alpha} \approx 10$. We estimated the unknown value $X = \frac{\sigma^2}{\epsilon^2}$ in the upper bound (18) of the critical batch size by using $\delta = 0.5$, $b^\star = 2^5$ (see Figure 6), and $\underline{\alpha} \approx \overline{\alpha} \approx 10$ as follows:

$$b^\star \approx \frac{\overline{\alpha}\sigma^2}{\{\underline{\alpha} - \delta(1 - c)\overline{\alpha}\}\epsilon^2} \approx X, \text{ i.e., } X \approx 32.$$

Let us estimate the critical batch size using $X \approx 32$ and Theorem 3.3(iii). For example, when using Algorithm 1 with $c = 0.25$ for training ResNet-34 on the CIFAR-100 dataset, the upper bound of the critical batch size is

$$\frac{\overline{\alpha}}{\underline{\alpha} - \delta(1 - c)\overline{\alpha}}X \approx 51.2 \approx 2^6 = b^\star,$$

which implies that the estimated critical batch size 51.2 is close to the measured critical batch size $b^\star = 2^6 = 64$ in Figure 6.

## 5 Conclusion

This paper presented a convergence analysis of SGD using the Armijo line search for nonconvex optimization. The analysis indicates that the upper bound of the expectation of the squared norm of the full gradient becomes smaller as the number of steps and the batch size grow. Moreover, we investigated the relationship between the number of steps and the batch size needed for nonconvex optimization of SGD using the Armijo line search. We showed that the number of steps needed for nonconvex optimization is monotone decreasing and convex with respect to the batch size; i.e., the steps decrease in number as the batch size increases. We also showed that the SFO complexity needed for nonconvex optimization is convex with respect to the batch size and that there exists a critical batch size at which the SFO complexity is minimized. In addition, we gave an upper bound on the critical batch size and showed that it can be estimated by using some parameters. Finally, we provided numerical results that support our theoretical findings. We trained ResNets on the CIFAR-10 and CIFAR-100 datasets and MLP on the MNIST dataset and found that SGD using the Armijo line search decreases the number of steps as the batch size increases and that SGD using the Armijo line search and the critical batch size performs better than other optimizers for any batch size in the sense of minimizing the SFO complexities needed to achieve high training accuracies. Moreover, we showed that the batch sizes estimated from the upper bound of the critical batch size are close to those of the numerical results.

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

# A Appendix

## A.1 Proof of lemma 2.1

(i) Let $k \in \mathbb{N}$ and let $L_{B_k}$ be the Lipschitz constant of $\nabla f_{B_k}$. Lemma 1 in (Vaswani et al., 2019) is as follows:

$$
\begin{aligned}
&\forall f_{B_k} \colon \mathbb{R}^d \to \mathbb{R} \ \forall c \in (0,1) \ \forall \boldsymbol{\theta}_k \in \mathbb{R}^d \ \forall \overline{\alpha} > 0 \\
&\exists \alpha_k \in (0, \overline{\alpha}] \ (f_{B_k}(\boldsymbol{\theta}_k - \alpha_k \nabla f_{B_k}(\boldsymbol{\theta}_k)) \leq f_{B_k}(\boldsymbol{\theta}_k) - c\alpha_k \|\nabla f_{B_k}(\boldsymbol{\theta}_k)\|^2) \\
&\Rightarrow \min\left\{\frac{2(1-c)}{L_{B_k}}, \overline{\alpha}\right\} \leq \alpha_k.
\end{aligned}
\tag{21}
$$

The negative proposition of (21) is as follows:

$$
\begin{aligned}
&\exists f_{B_k} \colon \mathbb{R}^d \to \mathbb{R} \ \exists c \in (0,1) \ \exists \boldsymbol{\theta}_k \in \mathbb{R}^d \ \exists \overline{\alpha} > 0 \\
&\exists \alpha_k \in (0, \overline{\alpha}] \ (f_{B_k}(\boldsymbol{\theta}_k - \alpha_k \nabla f_{B_k}(\boldsymbol{\theta}_k)) \leq f_{B_k}(\boldsymbol{\theta}_k) - c\alpha_k \|\nabla f_{B_k}(\boldsymbol{\theta}_k)\|^2) \\
&\wedge \min\left\{\frac{2(1-c)}{L_{B_k}}, \overline{\alpha}\right\} > \alpha_k.
\end{aligned}
\tag{22}
$$

We will prove that (22) holds. Let $n = b = 1$, $d = 1$, $c = 0.1$, $\overline{\alpha} = 1$, and $f(\theta) = f_{B_k}(\theta) = \theta^2$. From $\nabla f(\theta) = 2\theta$, we have that $L_{B_k} = 2$. Since $\theta^* = 0$ is the global minimizer of $f$, we set $\theta_k \in \mathbb{R}$ such that $\theta_k \neq \theta^*$. The Armijo condition in this case is such that $(\theta_k - 2\alpha_k\theta_k)^2 \leq \theta_k^2 - c\alpha_k(2\theta_k)^2$, which is equivalent to $\alpha_k \leq 1 - c = 0.9$. Hence,

$$
\begin{aligned}
&\exists \alpha_k \in (0,1] \ (\alpha_k \leq 0.9) \wedge (\min\{0.9, 1\} > \alpha_k) \\
&\Leftrightarrow \exists \alpha_k \in (0, \overline{\alpha}] \ (\alpha_k \leq 1 - c) \wedge \left(\min\left\{\frac{2(1-c)}{L_{B_k}}, \overline{\alpha}\right\} > \alpha_k\right) \\
&\Leftrightarrow \exists \alpha_k \in (0, \overline{\alpha}] \ (f_{B_k}(\theta_k - \alpha_k \nabla f_{B_k}(\theta_k)) \leq f_{B_k}(\theta_k) - c\alpha_k \|\nabla f_{B_k}(\theta_k)\|^2) \\
&\wedge \min\left\{\frac{2(1-c)}{L_{B_k}}, \overline{\alpha}\right\} > \alpha_k,
\end{aligned}
$$

which implies that (22) holds.

(ii) Since $\frac{\alpha_k}{\delta}$ does not satisfy the Armijo condition (11), we have that

$$
f_{B_k}\left(\boldsymbol{\theta}_k - \frac{\alpha_k}{\delta}\nabla f_{B_k}(\boldsymbol{\theta}_k)\right) > f_{B_k}(\boldsymbol{\theta}_k) - c\frac{\alpha_k}{\delta}\|\nabla f_{B_k}(\boldsymbol{\theta}_k)\|^2.
\tag{23}
$$

The $L_{B_k}$–smoothness of $f_{B_k}$ ensures that the descent lemma is true, i.e.,

$$
\begin{aligned}
&f_{B_k}\left(\boldsymbol{\theta}_k - \frac{\alpha_k}{\delta}\nabla f_{B_k}(\boldsymbol{\theta}_k)\right) \\
&\leq f_{B_k}(\boldsymbol{\theta}_k) + \left\langle \nabla f_{B_k}(\boldsymbol{\theta}_k), \left(\boldsymbol{\theta}_k - \frac{\alpha_k}{\delta}\nabla f_{B_k}(\boldsymbol{\theta}_k)\right) - \boldsymbol{\theta}_k \right\rangle + \frac{L_{B_k}}{2}\left\|\left(\boldsymbol{\theta}_k - \frac{\alpha_k}{\delta}\nabla f_{B_k}(\boldsymbol{\theta}_k)\right) - \boldsymbol{\theta}_k\right\|^2,
\end{aligned}
$$

which implies that

$$
f_{B_k}\left(\boldsymbol{\theta}_k - \frac{\alpha_k}{\delta}\nabla f_{B_k}(\boldsymbol{\theta}_k)\right) \leq f_{B_k}(\boldsymbol{\theta}_k) + \frac{\alpha_k}{\delta}\left(\frac{L_{B_k}\alpha_k}{2\delta} - 1\right)\|\nabla f_{B_k}(\boldsymbol{\theta}_k)\|^2.
\tag{24}
$$

Hence, (23) and (24) imply that

$$
-c\frac{\alpha_k}{\delta}\|\nabla f_{B_k}(\boldsymbol{\theta}_k)\|^2 \leq \frac{\alpha_k}{\delta}\left(\frac{L_{B_k}\alpha_k}{2\delta} - 1\right)\|\nabla f_{B_k}(\boldsymbol{\theta}_k)\|^2,
$$

which in turn implies that

$$
\frac{\alpha_k}{\delta}\left(\frac{L_{B_k}\alpha_k}{2\delta} - (1 - c)\right)\|\nabla f_{B_k}(\boldsymbol{\theta}_k)\|^2 \geq 0.
$$

Accordingly,

$$\frac{L_{B_k}\alpha_k}{2\delta} - (1-c) \geq 0, \text{ i.e., } \alpha_k \geq \frac{2\delta(1-c)}{L_{B_k}}.$$

From $L_{B_k} = \frac{1}{b}\sum_{i\in[b]} L_{\xi_{k,i}} \leq L := \max_{i\in[n]} L_i$ $(k \in \mathbb{N})$, we also have that $\alpha_k \geq \frac{2\delta(1-c)}{L_{B_k}} \geq \frac{2\delta(1-c)}{L}$.

## A.2 Proof of Theorem 3.1

The definition of $f(\boldsymbol{\theta}) := \frac{1}{n}\sum_{i\in[n]} f_i(\boldsymbol{\theta})$ and the $L_i$–smoothness of $f_i$ $(i \in [n])$ imply that, for all $\boldsymbol{\theta}_1, \boldsymbol{\theta}_2 \in \mathbb{R}^d$,

$$\|\nabla f(\boldsymbol{\theta}_1) - \nabla f(\boldsymbol{\theta}_2)\| \leq \frac{1}{n}\sum_{i\in[n]} \|\nabla f_i(\boldsymbol{\theta}_1) - \nabla f_i(\boldsymbol{\theta}_2)\| \leq \frac{\sum_{i\in[n]} L_i}{n}\|\boldsymbol{\theta}_1 - \boldsymbol{\theta}_2\|,$$

which in turn implies that $\nabla f$ is Lipschitz continuous with Lipschitz constant $L_n := \frac{1}{n}\sum_{i\in[n]} L_i$. Hence, the descent lemma ensures that, for all $k \in \mathbb{N}$,

$$f(\boldsymbol{\theta}_{k+1}) \leq f(\boldsymbol{\theta}_k) + \langle \nabla f(\boldsymbol{\theta}_k), \boldsymbol{\theta}_{k+1} - \boldsymbol{\theta}_k \rangle + \frac{L_n}{2}\|\boldsymbol{\theta}_{k+1} - \boldsymbol{\theta}_k\|^2,$$

which, together with $\boldsymbol{\theta}_{k+1} := \boldsymbol{\theta}_k - \alpha_k \nabla f_{B_k}(\boldsymbol{\theta}_k)$, implies that

$$f(\boldsymbol{\theta}_{k+1}) \leq f(\boldsymbol{\theta}_k) - \alpha_k \langle \nabla f(\boldsymbol{\theta}_k), \nabla f_{B_k}(\boldsymbol{\theta}_k) \rangle + \frac{L_n\alpha_k^2}{2}\|\nabla f_{B_k}(\boldsymbol{\theta}_k)\|^2. \tag{25}$$

From $\langle \boldsymbol{x}, \boldsymbol{y} \rangle = \frac{1}{2}(\|\boldsymbol{x}\|^2 + \|\boldsymbol{y}\|^2 - \|\boldsymbol{x} - \boldsymbol{y}\|^2)$ $(\boldsymbol{x}, \boldsymbol{y} \in \mathbb{R}^d)$, we have that, for all $k \in \mathbb{N}$,

$$\langle \nabla f(\boldsymbol{\theta}_k), \nabla f_{B_k}(\boldsymbol{\theta}_k) \rangle = \frac{1}{2}\left(\|\nabla f(\boldsymbol{\theta}_k)\|^2 + \|\nabla f_{B_k}(\boldsymbol{\theta}_k)\|^2 - \|\nabla f(\boldsymbol{\theta}_k) - \nabla f_{B_k}(\boldsymbol{\theta}_k)\|^2\right).$$

Accordingly, (25) implies that, for all $k \in \mathbb{N}$,

$$f(\boldsymbol{\theta}_{k+1}) \leq f(\boldsymbol{\theta}_k) - \frac{\alpha_k}{2}\left(\|\nabla f(\boldsymbol{\theta}_k)\|^2 + \|\nabla f_{B_k}(\boldsymbol{\theta}_k)\|^2 - \|\nabla f(\boldsymbol{\theta}_k) - \nabla f_{B_k}(\boldsymbol{\theta}_k)\|^2\right) + \frac{L_n\alpha_k^2}{2}\|\nabla f_{B_k}(\boldsymbol{\theta}_k)\|^2$$

$$= f(\boldsymbol{\theta}_k) - \frac{\alpha_k}{2}\|\nabla f(\boldsymbol{\theta}_k)\|^2 + \frac{1}{2}(L_n\alpha_k - 1)\alpha_k\|\nabla f_{B_k}(\boldsymbol{\theta}_k)\|^2 + \frac{\alpha_k}{2}\|\nabla f(\boldsymbol{\theta}_k) - \nabla f_{B_k}(\boldsymbol{\theta}_k)\|^2.$$

From $0 < \underline{\alpha} \leq \alpha_k \leq \overline{\alpha} < \frac{1}{L_n}$, we have that, for all $k \in \mathbb{N}$,

$$(L_n\alpha_k - 1)\alpha_k \leq (L_n\overline{\alpha} - 1)\alpha_k \leq (L_n\overline{\alpha} - 1)\underline{\alpha} < 0.$$

Hence, for all $k \in \mathbb{N}$,

$$f(\boldsymbol{\theta}_{k+1}) \leq f(\boldsymbol{\theta}_k) - \frac{\underline{\alpha}}{2}\|\nabla f(\boldsymbol{\theta}_k)\|^2 + \frac{1}{2}(L_n\overline{\alpha} - 1)\underline{\alpha}\|\nabla f_{B_k}(\boldsymbol{\theta}_k)\|^2 + \frac{\overline{\alpha}}{2}\|\nabla f(\boldsymbol{\theta}_k) - \nabla f_{B_k}(\boldsymbol{\theta}_k)\|^2. \tag{26}$$

Assumption 2.1 guarantees that

$$\mathbb{E}\left[\nabla f_{B_k}(\boldsymbol{\theta}_k)|\boldsymbol{\theta}_k\right] = \nabla f(\boldsymbol{\theta}_k) \text{ and } \mathbb{E}\left[\|\nabla f_{B_k}(\boldsymbol{\theta}_k) - \nabla f(\boldsymbol{\theta}_k)\|^2|\boldsymbol{\theta}_k\right] \leq \frac{\sigma^2}{b}. \tag{27}$$

Hence, we have

$$\mathbb{E}\left[\|\nabla f_{B_k}(\boldsymbol{\theta}_k)\|^2|\boldsymbol{\theta}_k\right]$$
$$= \mathbb{E}\left[\|\nabla f_{B_k}(\boldsymbol{\theta}_k) - \nabla f(\boldsymbol{\theta}_k) + \nabla f(\boldsymbol{\theta}_k)\|^2|\boldsymbol{\theta}_k\right]$$
$$= \mathbb{E}\left[\|\nabla f_{B_k}(\boldsymbol{\theta}_k) - \nabla f(\boldsymbol{\theta}_k)\|^2|\boldsymbol{\theta}_k\right] + 2\mathbb{E}\left[\langle \nabla f_{B_k}(\boldsymbol{\theta}_k) - \nabla f(\boldsymbol{\theta}_k), \nabla f(\boldsymbol{\theta}_k)\rangle|\boldsymbol{\theta}_k\right] + \mathbb{E}\left[\|\nabla f(\boldsymbol{\theta}_k)\|^2|\boldsymbol{\theta}_k\right] \tag{28}$$
$$\geq \|\nabla f(\boldsymbol{\theta}_k)\|^2.$$

Inequalities (26), (27), and (28) guarantee that, for all $k \in \mathbb{N}$,

$$\mathbb{E}\left[f(\boldsymbol{\theta}_{k+1})|\boldsymbol{\theta}_k\right] \leq f(\boldsymbol{\theta}_k) - \frac{\alpha}{2}\|\nabla f(\boldsymbol{\theta}_k)\|^2 + \frac{1}{2}(L_n\overline{\alpha} - 1)\underline{\alpha}\|\nabla f(\boldsymbol{\theta}_k)\|^2 + \frac{\overline{\alpha}\sigma^2}{2b}. \tag{29}$$

Taking the total expectation on both sides of (29) thus ensures that, for all $k \in \mathbb{N}$,

$$\frac{1}{2}\left\{\underline{\alpha} - (L_n\overline{\alpha} - 1)\underline{\alpha}\right\}\mathbb{E}\left[\|\nabla f(\boldsymbol{\theta}_k)\|^2\right] \leq \mathbb{E}\left[f(\boldsymbol{\theta}_k) - f(\boldsymbol{\theta}_{k+1})\right] + \frac{\overline{\alpha}\sigma^2}{2b}. \tag{30}$$

Let $K \geq 1$. Summing (30) from $k = 0$ to $k = K - 1$ ensures that

$$\frac{(2 - L_n\overline{\alpha})\underline{\alpha}}{2}\sum_{k=0}^{K-1}\mathbb{E}\left[\|\nabla f(\boldsymbol{\theta}_k)\|^2\right] \leq \mathbb{E}\left[f(\boldsymbol{\theta}_0) - f(\boldsymbol{\theta}_K)\right] + \frac{\overline{\alpha}\sigma^2 K}{2b},$$

which, together with the boundedness of $f$, i.e., $f_* \leq f(\boldsymbol{\theta}_k)$, implies that

$$\frac{(2 - L_n\overline{\alpha})\underline{\alpha}}{2}\sum_{k=0}^{K-1}\mathbb{E}\left[\|\nabla f(\boldsymbol{\theta}_k)\|^2\right] \leq \mathbb{E}\left[f(\boldsymbol{\theta}_0) - f_*\right] + \frac{\overline{\alpha}\sigma^2 K}{2b}.$$

Therefore, we have

$$\frac{1}{K}\sum_{k=0}^{K-1}\mathbb{E}\left[\|\nabla f(\boldsymbol{\theta}_k)\|^2\right] \leq \frac{2(f(\boldsymbol{\theta}_0) - f_*)}{(2 - L_n\overline{\alpha})\underline{\alpha}K} + \frac{\overline{\alpha}\sigma^2}{(2 - L_n\overline{\alpha})\underline{\alpha}b}.$$

Moreover, since we have

$$\min_{k \in [0:K-1]}\mathbb{E}\left[\|\nabla f(\boldsymbol{\theta}_k)\|^2\right] \leq \frac{1}{K}\sum_{k=0}^{K-1}\mathbb{E}\left[\|\nabla f(\boldsymbol{\theta}_k)\|^2\right],$$

the assertion in Theorem 3.1 holds.

### A.3  Proof of Theorem 3.2

(i) We have

$$\frac{C_1}{K} + \frac{C_2}{b} = \epsilon^2$$

is equivalent to

$$K = K(b) = \frac{C_1 b}{\epsilon^2 b - C_2}.$$

Hence, Theorem 3.1 leads to an $\epsilon$–approximation.

(ii) We have

$$\frac{\mathrm{d}K(b)}{\mathrm{d}b} = \frac{-C_1 C_2}{(\epsilon^2 b - C_2)^2} \leq 0 \text{ and } \frac{\mathrm{d}^2 K(b)}{\mathrm{d}b^2} = \frac{2C_1 C_2 \epsilon^2}{(\epsilon^2 b - C_2)^3} \geq 0,$$

which implies that $K$ is monotone decreasing and convex with respect to $b$.

### A.4  Proof of Theorem 3.3

(i) From

$$N(b) = \frac{C_1 b^2}{\epsilon^2 b - C_2},$$

we have

$$\frac{\mathrm{d}N(b)}{\mathrm{d}b} = \frac{C_1 b(\epsilon^2 b - 2C_2)}{(\epsilon^2 b - C_2)^2} \text{ and } \frac{\mathrm{d}^2 N(b)}{\mathrm{d}b^2} = \frac{2C_1 C_2^2}{(\epsilon^2 b - C_2)^3} \geq 0,$$

which implies that $N$ is convex with respect to $b$.

(ii) We have

$$\frac{\mathrm{d}N(b)}{\mathrm{d}b} \begin{cases} < 0 & \text{if } b < b^\star, \\ = 0 & \text{if } b = b^\star = \frac{2C_2}{\epsilon^2}, \\ > 0 & \text{if } b > b^\star. \end{cases}$$

Hence, the point $b^\star$ minimizes $N$.

(iii) Lemma 2.1(iii) ensures that

$$L_n := \frac{1}{n} \sum_{i \in [n]} L_i \leq L = \frac{2\delta(1-c)}{\underline{\alpha}}.$$

Hence,

$$b^\star = \frac{2C_2}{\epsilon^2} = \frac{2\overline{\alpha}\sigma^2}{(2 - L_n\overline{\alpha})\underline{\alpha}\epsilon^2} \leq \frac{2\overline{\alpha}\sigma^2}{\underline{\alpha}\epsilon^2} \frac{\underline{\alpha}}{2\{\underline{\alpha} - \delta(1-c)\overline{\alpha}\}} = \frac{\overline{\alpha}\sigma^2}{\{\underline{\alpha} - \delta(1-c)\overline{\alpha}\}\epsilon^2}.$$

## A.5 Proof of (12)

Let $K \geq 1$. From (25) and $\alpha_k := \alpha > 0$, we have that, for all $k \in \mathbb{N}$,

$$f(\boldsymbol{\theta}_{k+1}) \leq f(\boldsymbol{\theta}_k) - \alpha\langle\nabla f(\boldsymbol{\theta}_k), \nabla f_{B_k}(\boldsymbol{\theta}_k)\rangle + \frac{L_n\alpha^2}{2}\|\nabla f_{B_k}(\boldsymbol{\theta}_k)\|^2.$$

Hence, (27) and (28) ensure that, for all $k \in \mathbb{N}$,

$$\mathbb{E}\left[f(\boldsymbol{\theta}_{k+1})\right] \leq \mathbb{E}\left[f(\boldsymbol{\theta}_k)\right] - \alpha\mathbb{E}\left[\|\nabla f(\boldsymbol{\theta}_k)\|^2\right] + \frac{L_n\alpha^2}{2}\left(\mathbb{E}\left[\|\nabla f(\boldsymbol{\theta}_k)\|^2\right] + \frac{\sigma^2}{b}\right),$$

which implies that, for all $k \in \mathbb{N}$,

$$\alpha\left(1 - \frac{L_n\alpha}{2}\right)\mathbb{E}\left[\|\nabla f(\boldsymbol{\theta}_k)\|^2\right] \leq \mathbb{E}\left[f(\boldsymbol{\theta}_k) - f(\boldsymbol{\theta}_{k+1})\right] + \frac{L_n\alpha^2\sigma^2}{2b}.$$

Summing the above inequalities from $k = 0$ to $k = K - 1$ ensures that

$$\alpha\left(1 - \frac{L_n\alpha}{2}\right)\sum_{k=0}^{K-1}\mathbb{E}\left[\|\nabla f(\boldsymbol{\theta}_k)\|^2\right] \leq \mathbb{E}\left[f(\boldsymbol{\theta}_0) - f(\boldsymbol{\theta}_K)\right] + \frac{L_n\alpha^2\sigma^2 K}{2b}.$$

Since $f$ is bounded below by $f_* := \frac{1}{n}\sum_{i \in [n]} f_{i,*}$, we have

$$\min_{k \in [0:K-1]}\mathbb{E}\left[\|\nabla f(\boldsymbol{\theta}_k)\|^2\right] \leq \frac{1}{K}\sum_{k=0}^{K-1}\mathbb{E}\left[\|\nabla f(\boldsymbol{\theta}_k)\|^2\right] \leq \frac{2\mathbb{E}\left[f(\boldsymbol{\theta}_0) - f_*\right]}{\alpha(2 - L_n\alpha)K} + \frac{L_n\alpha\sigma^2}{(2 - L_n\alpha)b}.$$

