# OpenReview forum: "Relationship between Batch Size and Number of Steps Needed for Nonconvex Optimization of Stochastic Gradient Descent using Armijo Line Search"
_TMLR — Rejected by TMLR_

### Review · Reviewer_sQvN · 2023-08-15

**Summary Of Contributions:**

 This study analyzes SGD with an Armijo line search for nonconvex optimization. The analysis suggests that larger step numbers and batch sizes lead to smaller full gradient squared norms. The study also shows that for SGD with Armijo-line-search rates, more steps are needed for nonconvex optimization with smaller batch sizes. Stochastic first-order oracle (SFO) complexity, the cost of stochastic gradient computation, decreases with larger batch sizes. Numerical results support these findings, indicating that more substantial batch sizes require fewer steps to train neural networks, and there are critical batch sizes as per theoretical predictions.

**Audience:**

Yes

**Claims And Evidence:**

Yes

**Requested Changes:**

N/A

**Strengths And Weaknesses:**

In general, I am not very familiar with this topic.

I only have two comments.

1. I understand your experiments are trying to demonstrate the correctness of theoretical results. However, I do not find any baselines, which means your approach's effectiveness is unclear.  As you mentioned in the first paragraph of Section 1.1, there are many variants of SGD. However, I do not find any of them used as a baseline in the experiment.

2. The experiments only considered two very relatively small neural networks. I would like to see the method performance using large networks such as wide-resnet.

---

> ### Author Response · Authors · 2023-09-21
> **Replies to Reviewer sQvN's comments**
>
> We would like to express our gratitude to the Action Editor and the reviewers for their valuable comments on our manuscript. We appreciate their detailed assessments and helpful feedback. We have revised the manuscript to incorporate all of the recommendations, which has resulted in an improved presentation of our work. The revised parts of the manuscript are marked in red.
>
> **Main Comments:**
> 1. I understand your experiments are trying to demonstrate the correctness of theoretical results. However, I do not find any baselines, which means your approach's effectiveness is unclear. As you mentioned in the first paragraph of Section 1.1, there are many variants of SGD. However, I do not find any of them used as a baseline in the experiment.
>
> 2. The experiments only considered two very relatively small neural networks. I would like to see the method performance using large networks such as wide-resnet.
>
> **Reply:** Thank you for your valuable comments. The revised manuscript gives numerical comparisons of SGD using the Armijo line search (SGD+Armijo) with the existing optimizers, such as SGD using a constant learning rate, the momentum method, Adam, AdamW, and RMSProp for training ResNet-34 on the CIFAR-10 and CIFAR-100 datasets, MLP on the MNIST dataset, and Wide ResNet-50-2 on the CIFAR-10 dataset. The numerical results show that SGD+Armijo using the critical batch size performs better than the other optimizers for any batch size in the sense of minimizing the SFO complexities needed for achieving high training accuracies. Please see Section 4 in the revised manuscript.

---

### Review · Reviewer_FGry · 2023-08-18

**Summary Of Contributions:**

The paper considers the classical SGD method with the backtracking Armijo-line-search method. The authors provide the theorem that ensures the convergence of this method. They also analyze how the convergence rate depends on batch size and obtain an optimal one that minimizes a SFO complexity. Finally, the theoretical results are supported by experiments.

**Audience:**

Yes

**Broader Impact Concerns:**

.

**Claims And Evidence:**

No

**Requested Changes:**

.

**Strengths And Weaknesses:**

The considered problem is very important for modern optimization tasks since one of the biggest problems in SGD is step size fine-tuning. The motivation of the paper is clear. However, looking through the paper I didn't find the current proofs convincing. Let me point to the main issues in the proof:

At the beginning of page 15, the authors assume that $f(\theta_k) - f(\theta_{k+1}) \geq 0.$ Using this assumption, they use the inequality
$$E[\frac{f(\theta_k) - f(\theta_{k+1})}{\alpha_k}] \leq \frac{L}{2 \delta (1 - c)}E[f(\theta_k) - f(\theta_{k+1})] ( ** ),$$ because $\alpha_k \geq \frac{2 \delta (1 - c)}{L}.$ Unfortunattely, we can not do this because $E[\cdot]$ is the **total** expectation. They can only use this inequality with the conditional expectation $E[\cdot | f(\theta_k) - f(\theta_{k+1}) \geq 0].$ So, this it true that
$$E[\frac{f(\theta_k) - f(\theta_{k+1})}{\alpha_k}| f(\theta_k) - f(\theta_{k+1}) \geq 0] \leq \frac{L}{2 \delta (1 - c)}E[f(\theta_k) - f(\theta_{k+1})| f(\theta_k) - f(\theta_{k+1}) \geq 0],$$
while ( ** ) is not true in general (at least, it is not proved). The same problem is in the other places on page 15. So it is required a major revision of the proof with the conditional expectation. For me, now it is not clear that this issue won't break anything in the other places.

Let me point to other weaknesses:

1. It seems that the main result of the paper is Theorem 3.1. From Theorem 3.1, Table 1, or the discussion at the end of page 3, it is not clear which method converges faster: SGD with constant step size or SGD with backtracking. In Table 1, is it true that $N$ in the "Constant $\alpha$" line is always larger (up to a constant factor) than $N$ from the "Armijo" line? Can we find some regimes when the latter is strongly better than the former?
2. $C(\theta_0, f_*)$ depends on $E[f(\theta_{k_0 + 1})].$ Can we find a reasonable bound for it? Otherwise, we can not compare Armijo's rule with the constant step size.

Minor:
1. What is $\Xi$ in Section 2.2? Is it finite or not? Can it be uncountable, for instance?
2. If we take $b = 1,$ is it well know that SGD converges to $\varepsilon$-stationary point with an appropriate step size. In Theorem 3.1, we can not take $b = 1$ in general to ensure convergence. This theorem works only with $b \approx \sigma^2 / \varepsilon.$ Is it true that it is difficult to prove something with Armijo's rule and small batch size?
3. Can you list all considered assumptions in Theorem 3.1? (not only Assumption 2.1)

Minor-Minor:
1. Can the authors add the "hyperref" package to their paper's source file? Without it, it is not very easy to "jump" between theorems, lemmas, assumptions, and formulas in the corresponding pdf file.

---

> ### Author Response · Authors · 2023-09-21
> **Replies to Reviewer FGry's comments**
>
> We would like to express our gratitude to the Action Editor and the reviewers for their valuable comments on our manuscript. We appreciate their detailed assessments and helpful feedback. We have revised the manuscript to incorporate all of the recommendations, which has resulted in an improved presentation of our work. The revised parts of the manuscript are marked in red.
>
> **Main Comment 1:**
> However, looking through the paper I didn't find the current proofs convincing.
>
> **Reply:** Thank you very much for pointing out. We revised the proof of Theorem 3.1. Please see Appendix A.2 for details. During revision of the proof of Theorem 3.1, we found that we could delete $C(\theta_0,f_*)$.
>
> **Main Comment 2:**
> Other weaknesses
>
> **Reply:** Thank you for your valuable comments. We compare the SFO complexity $N_{\mathrm{C}}$ for SGD using a constant learning rate $\alpha$ with $N_{\mathrm{A}}$ for SGD using the Armijo-line-search learning rate $\alpha_k$ $(< \overline{\alpha})$. Let $C_{1,\mathrm{C}}$ (resp. $C_{2,\mathrm{C}}$) be $C_1$ (resp. $C_{2}$) in Table 1 for SGD using a constant learning rate and let $C_{1,\mathrm{A}}$ (resp. $C_{2,\mathrm{A}}$) be $C_1$ (resp. $C_2$) in Table 1 for SGD using the Armijo-line-search learning rate. We have that
> \begin{align}
> &C_{1,\mathrm{A}} < C_{1,\mathrm{C}}
> \text{ iff }
> \overline{\alpha} < \frac{2}{L_n} - \frac{(t+1)(2-L_n \alpha)\alpha L}{2 \delta (1-c) L_n} \text{ }
> \left(< \frac{2}{L_n} \right),
> \end{align}
> \begin{align}
> &C_{2,\mathrm{A}} < C_{2,\mathrm{C}}
> \text{ iff }
> \overline{\alpha} < \frac{\sigma_{\mathrm{C}}^2 (2 - L_n \overline{\alpha})}{\sigma_{\mathrm{A}}^2 (2 - L_n \alpha)} \alpha
> \text{ }
> \bigg(
> \approx
> \frac{\sigma_{\mathrm{C}}^2}{\sigma_{\mathrm{A}}^2}  \alpha
> \bigg),
> \end{align}
> where $\sigma_{\mathrm{C}}^2$ (resp. $\sigma_{\mathrm{A}}^2$) denotes the upper bound of the variance of the stochastic gradient for SGD using a constant learning rate $\alpha$ (resp. the Armijo-line-search learning rate). Hence, if the above inequalities (A) hold, then SGD using the Armijo-line-search learning rate converges faster than SGD using a constant learning rate in the sense that
> \begin{align}
> \frac{C_{1,\mathrm{A}} b^2}{\epsilon^2 b - C_{2,\mathrm{A}}} = N_{\mathrm{A}} < N_{\mathrm{C}}
> = \frac{C_{1,\mathrm{C}} b^2}{\epsilon^2 b - C_{2,\mathrm{C}}}.
> \end{align}
> It would be difficult to check exactly that (A) holds before implementing SGD, since (A) involves unknown parameters, such as $L_n = \frac{1}{n} \sum_{i\in [n]} L_i$, $\sigma_{\mathrm{C}}^2$, and $\sigma_{\mathrm{A}}^2$. However, it can be expected that (A) holds, since it is known empirically that the relationship between the Armijo-line-search learning rate $\alpha_k$ and a constant learning rate $\alpha$ is $\alpha < \alpha_k$ $(< \overline{\alpha} < \frac{2}{L_n})$ (Section 3.3 provides the derivation of condition (A)).
>
> To verify whether SGD using the Armijo-line-search learning rate performs better than SGD using a constant learning rate (see the discussion in condition (A), we numerically compared SGD using the Armijo-line-search learning rate with not only SGD using a constant learning rate but also variants of SGD, such as the momentum method, Adam, AdamW, and RMSProp. We found that SGD using the Armijo-line-search learning rate and the critical batch size performs better than the other optimizers in the sense of minimizing the SFO complexities needed to achieve high training accuracies. Please see Section 4 in the revised manuscript.
>
> **Reply to Minor 1:** We think that $\Xi$ may be finite/infinite.
>
> **Reply to Minor 2:** We consider setting small batch sizes, e.g., $b = 1$. From the condition of the domain of $K$, we need to satisfy
> \begin{align}
> b = 1 > \frac{C_2}{\epsilon^2} \text{ iff }
> \overline{\alpha} < \frac{2 \epsilon^2}{L_n (\sigma^2 + \epsilon^2)} \text{ }
> \left(< \frac{2}{L_n} \right)
> \end{align}
> to ensure the results in Theorem 3.2. If the upper bound $\overline{\alpha}$ satisfies the more restricted condition than $\overline{\alpha} < \frac{2}{L_n}$, then SGD using the Armijo line search with $b=1$ and $K(1) = \frac{C_1}{\epsilon^2 - C_2}$ is an $\epsilon$--approximation. Please see also Section 3.2 in the revised manuscript.
>
> **Reply to Minor 3:** The list of all considered assumptions in Theorem 3.1 is in Section 3.1. Please check it.
>
> **Reply to Minor-Minor:** Thank you for your comment. We added the ``hyperref" package. Please check the revised manuscript.

---

> > ### Comment · Reviewer_FGry · 2023-09-26
> > **Reply**
> >
> > > Main Comment 1: However, looking through the paper I didn't find the current proofs convincing.
> >
> > > Reply: Thank you very much for pointing out. We revised the proof of Theorem 3.1. Please see Appendix A.2 for details. During revision of the proof of Theorem 3.1, we found that we could delete
> >
> > The problems are still in the proof. Let me point to one of them:
> > On page 19, after eq. (30), the authors bound $A_k:$
> > $$A_k := E[\sum_{k=0}^{K-1} \frac{f(\theta_k) - f(\theta_{k+1})}{\alpha_k}] \leq \sum_{k=0}^{K-1} \frac{f(\theta_k) - f(\theta_{k+1})}{\underline{\alpha}} + \sum_{k \in S_K} \left(\frac{f(\theta_k) - f(\theta_{k+1})}{\bar{\alpha}} - \frac{f(\theta_k) - f(\theta_{k+1})}{\underline{\alpha}}\right).$$
> >
> > The last inequality is not correct because $E[\cdot]$ has disappeared! Note that $S_K$ is a **random** set.
> >
> > Another major problem in the dependence on $t.$ Again, $t$ is a **random** variable! The authors provide the convergence rate that depends on the random variable $t$ in Theorem 3.1. Also, $t$ can be as large as $K,$ so Theorem 3.1 does not guarantee the improvement with the number of iterations $K.$
> >
> > Final decision: it is clear that the paper has major mathematical flaws that should be fixed. Right now, I recommend the rejection. My recommendation: I suggest the authors to revisit the proofs and **carefully** look through them. I know that the problem of combining the adaptivity of step sizes with SGD is a very challenging task; there have been many attempts in the past. However, as far as I know, none of the previous works could do it without some additional assumptions.

---

> ### Author Response · Authors · 2023-10-05
> **Reply to Reviewer FGry's comments**
>
> Thank you very much for pointing out. We revised the proof of Theorem 3.1.  Theorem 3.1 is as follows:
> \begin{align}
> \min_{k \in [0:K-1]} \mathbb{E}[ \Vert \nabla f(\theta_k) \Vert^2 ]
> \leq \frac{2(f( \theta_{0}) - f_*)}{(2 - L_n \overline{\alpha}) \underline{\alpha}}
> \frac{1}{K} +
> \frac{\overline{\alpha} \sigma^2}{(2-L_n \overline{\alpha}) \underline{\alpha}} \frac{1}{b},
> \end{align}
> where $\delta, c \in (0,1)$, $L := \max_{i\in [n]} L_i$, $L_n := \frac{1}{n} \sum_{i\in [n]} L_i$ $(\leq L)$, $f_* := \frac{1}{n} \sum_{i\in [n]} f_{i,*}$, $\underline{\alpha} := \frac{2 \delta (1-c)}{L}$,
> and $\overline{\alpha} < \frac{1}{L_n}$.
>
> Please see Appendix A.2 for details. During revision of the proof of Theorem 3.1, we found that we could delete using $t$. The revised parts of the manuscript are marked in red.

---

### Review · Reviewer_yfru · 2023-09-30

**Summary Of Contributions:**

The paper considers the stochastic gradient descent with Armijo search. The main result is a convergence rate of SGD with minibatch for nonconvex problems under variance assumptions and smoothness assumptions. Based on this, the paper studies the number of steps to achieve a desired $\epsilon$-approximation and the critical batch size to achieve minimal SFO gradient complexity. Experimental results on MNIST, CIFAR-10, CIFAR-100 are presented which demonstrate the evolution of SFO complexity and the number of steps required (to reach a given accuracy) for both the presented method and some baseline methods such as ADAM, ADAMw, RMSProp, etc. The results on CIFAR with ResNet-34 demonstrate the method outperforms baselines when selecting the baseline according to the theoretical optimal value derived in the paper. The results on wide ResNet are less conclusive.

**Audience:**

Yes

**Claims And Evidence:**

Yes

**Requested Changes:**

Please address the errors/issues in the analysis: provide a sharp upper bound on the random variable $t$ to make Theorem 3.1 non-vacuous. Comment on how to ensure the condition $f(\theta_k)\leq f(\theta_0)$ can hold for all $k$.

**Strengths And Weaknesses:**

Strengths:

The paper considers SGD with minibatch and Armijo line search. This implementation does not require knowledge of the smoothness parameter, which can be difficult to estimate in practice. The convergence rate shows clearly how the batch size would benefit the convergence rate and provides a concrete recommendation for the optimal batch size.
The obtained bounds match the existing analysis of SGD with constant step size, even though the latter do require knowledge of the smoothness parameter.


========== Weaknesses===========

As far as I see, there seems to be a serious problem in the theoretical analysis. In particular, the bound in Theorem 3.1 is expressed in terms of the quantity $t$, which is the number of iterations where the function value increases. This number is a random variable, and is very complicated to estimate. The left-hand side of the inequality involves an expectation. Therefore, Theorem 3.1 gives a bound for a non-random variable in terms of a random variable which is very difficult to estimate in practice. A priori, the random variable $t$ could be as large as $K$ and in this case, the bound is completely vacuous.  Thus, the bound does not make much sense unless a separate argument is used to estimate $t$ (and show it to be small relative to $K$).  Furthermore, Theorem 3.1 requires an assumption $f(\theta_k)\leq f(\theta_0)$.  This assumption may not hold in practice and cannot be verified for any particular case without running the algorithm.



===================== Minor typos===========
Beginning of section 2.3.2: (called a learning rate in machine learning field)= > “(called a learning rate in the machine learning field)” or “(called a learning rate in machine learning)”

Above equation (12): “SGD with a constant learning rate satisfies that” = > “SGD with a constant learning rate satisfies:”

Page 11: “Various values c” = > “various values of c”

---

> ### Author Response · Authors · 2023-10-05
> **Reply to Reviewer yfru's comments**
>
> We would like to express our gratitude to the Action Editor and the reviewers for their valuable comments on our manuscript. We appreciate their detailed assessments and helpful feedback. We have revised the manuscript to incorporate all of the recommendations, which has resulted in an improved presentation of our work. The revised parts of the manuscript are marked in red.
>
> **Main Comment:**
> Please address the errors/issues in the analysis: provide a sharp upper bound on the random variable $t$ to make Theorem 3.1 non-vacuous. Comment on how to ensure the condition $f(\theta_k) \leq f(\theta_0)$ can hold for all $k$.
>
> **Reply:**
> Thank you very much for pointing out. We revised the proof of Theorem 3.1. Theorem 3.1 is as follows:
> \begin{align}
> \min_{k \in [0:K-1]} \mathbb{E}[ \Vert \nabla f(\theta_k) \Vert^2 ]
> \leq \frac{2(f( \theta_{0}) - f_*)}{(2 - L_n \overline{\alpha}) \underline{\alpha}}
> \frac{1}{K} +
> \frac{\overline{\alpha} \sigma^2}{(2-L_n \overline{\alpha}) \underline{\alpha}} \frac{1}{b},
> \end{align}
> where $\delta, c \in (0,1)$, $L := \max_{i\in [n]} L_i$, $L_n := \frac{1}{n} \sum_{i\in [n]} L_i$ $(\leq L)$, $f_* := \frac{1}{n} \sum_{i\in [n]} f_{i,*}$, $\underline{\alpha} := \frac{2 \delta (1-c)}{L}$,
> and $\overline{\alpha} < \frac{1}{L_n}$.
>
> Please see Appendix A.2 for details. During revision of the proof of Theorem 3.1, we found that we could delete $f(\theta_k) \leq f(\theta_0)$ and $t$.

---

### Decision · Action_Editor_ca14 · 2023-10-29

**Recommendation:** Reject

**Comment:**

The submission is not rigorous in some sense. The initial submission contains severe mistakes as identified by one of the reviewers. In particular, they take a condition to get a bound and ignore this condition in taking the whole expectation. The second submission tries to address the mistake, which, however, also introduces new issues. That is, the upper bound involves a very complicated random variable $t$, which can lead to vacuous bounds.

Regarding the convergence rates, the upper bound involves the parameter $L$, which is the maximum of smoothness parameters over all training examples. Note traditional convergence rate would imply a bound depending on $L_N$, which is the average of smoothness parameters over all training examples. In practice, $L$ can be much larger than $L_n$, and in this case, the derived bound is not appealing. Furthermore, the convergence rate is of the order $L\max(1/K,1/b)$, which means that the batchsize should be of the order $K$ to get the rate $L/K$. This batchsize is too large, and is not often used in practical implementations.

A key idea of the analysis is that the proposed step size has a lower bound. In the literature, there are other methods which have this property and do not need the information on the smoothness parameter. For example, one can decrease the step size until the descent property satisfies
$$
f_{B_k}(\theta_k-\alpha\nabla f_{B_k}(\theta_k))\leq \alpha/2\|\nabla f_{B_k}(\theta_k)\|.
$$
As shown in https://www.cs.ubc.ca/~schmidtm/Courses/5XX-S22/S1.pdf (page 22), this algorithm would always get a step size no smaller than 1/(2L). The discussion with the above method is not included in this submission.

**Audience:**

Stochastic optimization is a popular topic and the proposed algorithm does not require information on the smoothness parameter. Therefore, there should be some audiences interested in the work.

**Claims And Evidence:**

The paper studies minibatch stochastic gradient descent with the Armijo line search. The algorithm does not need the information on the smoothness parameter. It is shown the step size selected by Armijo line search has an upper and a lower bound. Based on this, the paper develops convergence rates in terms of the square norm of the gradients for nonconvex problems. Experimental results are also proposed.